# Health economic evaluation of strategies to eliminate *gambiense* human African trypanosomiasis in the Mandoul disease focus of Chad

**Marina Antillon**[1,2]*, **Ching-I Huang**[3,4], **Samuel A. Sutherland**[3,5], **Ronald E. Crump**[3,4], **Paul R. Bessell**[6], **Alexandra P. M. Shaw**[7,8], **Iñaki Tirados**[9], **Albert Picado**[10], **Sylvain Biéler**[10], **Paul E. Brown**[3,4], **Philippe Solano**[11], **Severin Mbainda**[12], **Justin Darnas**[12], **Xia Wang-Steverding**[3,5], **Emily H. Crowley**[3,4], **Mallaye Peka**[12]†, **Fabrizio Tediosi**[1,2], **Kat S. Rock**[3,4]

**1** Swiss Tropical and Public Health Institute, Basel, Switzerland, **2** University of Basel, Basel, Switzerland, **3** Zeeman Institute for Systems Biology and Infectious Disease Epidemiology Research, The University of Warwick, Coventry, United Kingdom, **4** Mathematics Institute, The University of Warwick, Coventry, United Kingdom, **5** Warwick Medical School, University of Warwick, Coventry, United Kingdom, **6** Independent Consultant, Edinburgh, United Kingdom, **7** Infection Medicine, Deanery of Biomedical Sciences, College of Medicine and Veterinary Medicine, The University of Edinburgh, Edinburgh, United Kingdom, **8** AP Consultants, Walworth Enterprise Centre, Andover, United Kingdom, **9** Department of Vector Biology, Liverpool School of Tropical Medicine, United Kingdom, **10** Foundation for Innovative New Diagnostics (FIND), Geneva, Switzerland, **11** Institut de Recherche pour le Développement, UMR INTERTRYP IRD-CIRAD, Université de Montpellier, Montpellier, France, **12** Programme National de Lutte contre la Trypanosomiase Humaine Africaine (PNLTHA), Moundou, Chad

† Deceased.
* marina.antillon@aya.yale.edu

**Data Availability Statement:** Information about the data used for fitting is described in Rock et al 2022 (Inf. Dis. of Pov). Screening data in this study

## Abstract

Human African trypanosomiasis, caused by the *gambiense* subspecies of *Trypanosoma brucei* (gHAT), is a deadly parasitic disease transmitted by tsetse. Partners worldwide have stepped up efforts to eliminate the disease, and the Chadian government has focused on the previously high-prevalence setting of Mandoul. In this study, we evaluate the economic efficiency of the intensified strategy that was put in place in 2014 aimed at interrupting the transmission of gHAT, and we make recommendations on the best way forward based on both epidemiological projections and cost-effectiveness. In our analysis, we use a dynamic transmission model fit to epidemiological data from Mandoul to evaluate the cost-effectiveness of combinations of active screening, improved passive screening (defined as an expansion of the number of health posts capable of screening for gHAT), and vector control activities (the deployment of Tiny Targets to control the tsetse vector). For cost-effectiveness analyses, our primary outcome is disease burden, denominated in disability-adjusted life-years (DALYs), and costs, denominated in 2020 US$. Although active and passive screening have enabled more rapid diagnosis and accessible treatment in Mandoul, the addition of vector control provided good value-for-money (at less than $750/DALY averted) which substantially increased the probability of reaching the 2030 elimination target for gHAT as set by the World Health Organization. Our transmission modelling and economic evaluation suggest that the gains that have been made could be maintained by passive

were used to inform future potential screening coverage and were obtained through the WHO HAT Atlas, which cannot be shared publicly but is under the stewardship of the WHO and available from the WHO (contact neglected.diseases@who.int or visit https://www.who.int/teams/control-of-neglected-tropical-diseases/human-african-trypanosomiasis/atlas-of-hat) for researchers who meet the criteria for access to confidential data. Assumptions and estimates were parameterized according to conventions in the economic evaluation literature. For assumptions around intervention, treatment effects and costs, see Table 2, S1 Text Section S1.3 and S4 text. Output data derived through the present study and the code to carry out all analyses are available from OSF https://osf.io/bjxwn/.

**Funding:** This work was supported by the Bill and Melinda Gates Foundation (www.gatesfoundation.org) through the Human African Trypanosomiasis Modelling and Economic Predictions for Policy (HAT MEPP) project [OPP1177824 and INV-005121] (MA, CH, SAS, REC, PEB, XWS, EHC, KSR, and FT) and the Trypa-NO! project [INV-008412 and INV-001785] (PRB, APMS, IT, AP, SB, and PS). The funders of the study did not have any role in the study design, data collection, data analysis, data interpretation, or the composition of the manuscript.

**Competing interests:** The authors have declared that no competing interests exist.

screening. Our analysis speaks to comparative efficiency, and it does not take into account all possible considerations; for instance, any cessation of ongoing active screening should first consider that substantial surveillance activities will be critical to verify the elimination of transmission and to protect against the possible importation of infection from neighbouring endemic foci.

## Author summary

In a drive to eliminate human African trypanosomiasis (gHAT or "sleeping sickness") from Chad following a peak in cases in 2002, the National Sleeping Sickness Control programme and its partners focused on making substantial changes to interventions within the high prevalence setting of Mandoul. These included the use of vector control starting in 2014 and improved screening in health facilities starting in 2015. To explore whether these interventions were an efficient use of resources we carried out a retrospective analysis using a dynamic transmission model fit to epidemiological data from Mandoul combined with a cost model. Our analysis indicated that improvements to passive screening enabled more rapid diagnosis and accessible treatment in Mandoul, and furthermore the addition of vector control was good value-for-money and substantially increased the probability of reaching the 2030 elimination of transmission target for gHAT set by the World Health Organization. Looking forwards, our prospective analysis also considers the health economics of future strategies and concludes that the scaleback of vertical interventions appears cost-effective if passive screening remains operational in Mandoul. This could therefore enable the shifting of resources to tackle other remaining foci in Chad.

## Introduction

In 1993, after forty years without any case reports of *gambiense* human African trypanosomiasis (gHAT) in the Mandoul region of Chad, healthcare personnel at the newly opened Catholic Mission Health Centre diagnosed a patient with gHAT, setting off new efforts against the disease in the region [1]. Commonly referred to as "sleeping sickness", the disease is caused by the *Trypanosoma brucei gambiense* parasite, and it is characterized by undifferentiated symptoms in the initial blood-borne stage and neurological symptoms in the second stage, which are generally fatal if untreated [2, 3]. In 1993, the first year of renewed reports in Mandoul, 212 cases were diagnosed in total, comprising most of the gHAT cases in Chad, followed by 2,291 more cases during the following fourteen years in that focus [1]. The detection of disease in the Mandoul focus paralleled an uptick in reported cases throughout the continent. The epidemic lasted more than a decade, during which 3,316 cases were reported between 1990 and 2004 in Chad [4] with 37,000 cases reported continent-wide at its peak in 1998. It is likely that many more infections went undiagnosed due to constraints that limited screening activities, presumably with untreated cases almost invariably culminating in the death of the infected person [5–7].

In many respects gHAT presents a number of unique challenges compared to most other infections targeted for control or elimination: there is no vaccine against the disease; the treatment, which causes a number of side-effects, cannot be used for mass-drug administration [8]; until 2020, a lumbar puncture for disease staging was required to assign the correct treatment to all cases [2]; after 2020, a lumbar puncture remains necessary for patients with signs of late

stage-2 disease; and control of the vector, the tsetse (*Glossina* spp.), requires alternative interventions to those popular for controlling mosquitoes [9]. Despite this, and after years of investment in screening campaigns, the appropriate diagnosis, and referral for treatment, the last decade has been characterized by optimism. In 2012 the World Health Organization (WHO) marked the disease for elimination as a public health problem by 2020 [10], and for the elimination of transmission by 2030 [11]. The 2020 goal was defined for the continent as a 90% reduction, compared to the 2000–2004 baseline, in areas at moderate or high risk (defined as more than one new reported case per year per 10,000 people averaged over 5 consecutive years) and fewer than 2000 annual cases globally [12]. The within-country indicator was given as less than one new reported case per year per 10,000 people for 5 years in each health district by 2020 [10]. The 2030 goal is defined as a complete cessation of transmission to humans by 2030, although the indicators for verification have not yet been published [13]. By 2017, there were fewer than 2000 reported cases continent-wide, outpacing the case reduction indicator set for the 2020 goal, although the reduction in areas at risk slightly missed the target (achieved in 83% of districts rather than 90%) [14].

Within Chad, there are three foci that are of concern: Maro and Moissala, close to the border of the Central African Republic, and Mandoul [15]. Two additional foci, Gore and Tapol, which have historically had cases, appear to no longer be active [16]. Starting in 2014, after a decade of case detection efforts, the number of cases showed a decreasing, but unremitting, low level of cases in the last years. Therefore, in that year, intensified screening in health facilities and a vector control project were put in place by the national gHAT control programme (Programme National de lutte contre la Trypanosomiase Humaine Africaine; PNLTHA-Chad) and international non-profit and academic partners [17–19]. Previous publications detail the impacts of these efforts on case reporting and transmission [20, 21]. The latest, Rock et al. [21], using mathematical transmission modeling and data from 2000–2019, strengthens and reiterates the conclusions that the last transmission event occurred in 2015 in Mandoul, although it was expected for the last case to be reported several years after this. Elimination as a public health problem has not yet been validated for Chad and must be done at a country level, rather than focus by focus [14]. No country has yet been verified as having achieved elimination of transmission and the criteria for this goal are yet to be made publicly available.

Here we will focus on the economic efficiency of strategies in Mandoul. Two other studies have looked at the costs of screening and vector control separately but not in concert, and so would miss the synergies possible in simultaneous interventions to halt transmission [19, 22]. We set out to perform a retrospective analysis for the Mandoul focus in which we look at what was done from 2014 compared to three strategies that would have omitted intervention components of the national programme's strategy. In this retrospective analysis, we also assessed the hypothetical scenario of whether the earlier availability of the oral drug, fexinidazole—which was actually rolled out in 2020—would have changed our assessment of which strategy had the highest probability of being cost-effective. Our subsequent prospective analysis also considers the health economic implications of what could be done against gHAT going forward from 2023 for Mandoul.

## Methods

### Setting and the historical control of gHAT

The focus of Mandoul (ca. 8.12˚N, 17.11˚E) spans 840 km$^2$. It contains no paved roads and is characterized by a gallery forest along a marshland. The focus spans five cantons—Bodo, Beboto, Dilingala, Koldaga, and Bekourou—across two provinces—Logone Oriental and

Mandoul (see Fig A in S1 Text) [19, 20]. The focus is bisected by Niaméte river on the southern part with temporary tributaries during the wet season from May to September [23]. Towards the north, the river gives way to marshland.

As of 2013, the focus was dotted by 114 settlements and contained about 38,674 people of the Sara ethnic group: 1,029 people lived in 22 encampments of fewer than 100 people, 17,626 lived in hamlets of between 100–500 people and 20,016 people lived in 27 villages of more than 500 people [20]. During the dry season, when the northern marshland dries out, it is populated by pastoral Bororo nomads [23]. The population lives off subsistence farming in traditional huts with no modern amenities. Throughout the focus, the marshy waterway is traversed via several cross-ways, used especially during market days when people on the eastern side of the focus cross to the west, where the hamlet of Bodo is located within Logone Oriental Province. In general, this focus is relatively isolated compared to Moissala and Maro; Moissala is contiguous with the Maitikolo focus of the Central African Republic (CAR) and often receives refugees from CAR while Maro sits along the CAR border in the east of Chad [24].

Screening, diagnosis, and treatment for gHAT in Chad is coordinated since the early 1990's by PNLTHA-Chad, which is headquartered in Moundou, about 150km west of the Mandoul focus [1, 23]. Until 2014, passive diagnosis of cases was only available in the Catholic Mission Hospital in the town of Bodo, which is the main service centre and transport hub of the focus (Fig 1). Active screening of at-risk villages was also conducted during the period but the temporal coverage of active screening varied contingent on available funding and security conditions.

Historically, PNLTHA-Chad has been funded with help from the *l'Organization de Coordination pour la Lutte contre les Endémies en Afrique centrale* (OCEAC) and the WHO, while diagnosis and treatment at the Catholic Mission Hospital is financed by user fees and private charity funds. Treatment medications have been donated via WHO to PNLTHA since 2002. In addition to the continued rounds of vehicle-based active screening that were funded by WHO, medical screening activities have been expanded since 2015 with funding from the

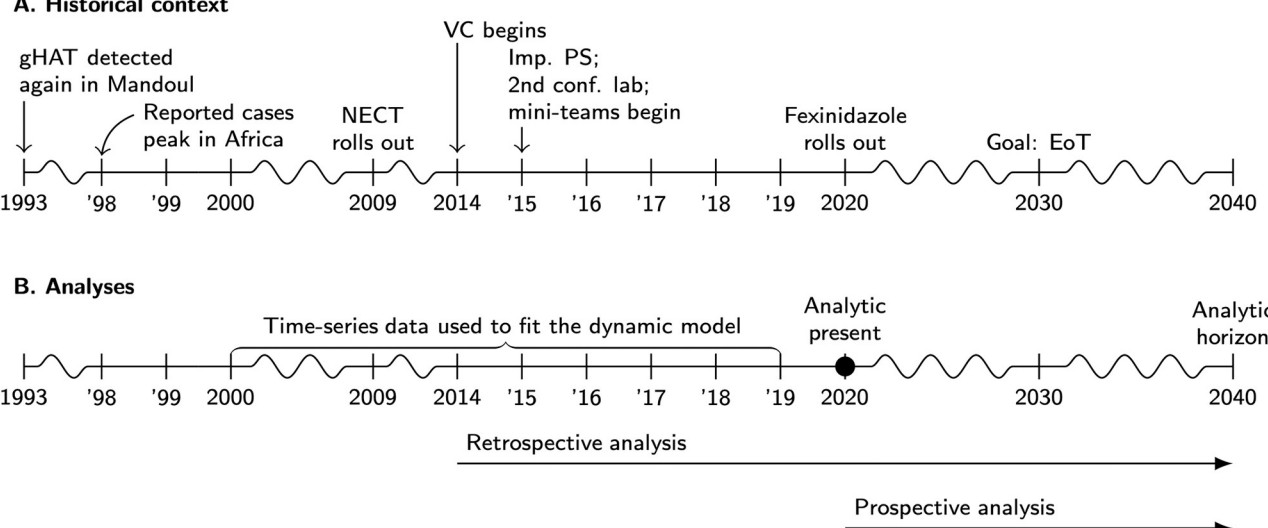

**Fig 1. Timeline of gHAT epidemiology and analytic frame.** A. Epidemiological information, start of new or improved interventions in Mandoul, and the WHO 2030 goal. B. The data that were used to fit the model and the years that correspond to our retrospective and prospective analysis. NECT: Nifurtimox-eflornithine combination therapy, VC: vector control, Imp. PS: Improved passive screening, EoT: elimination of transmission.

Swiss Agency for Development and Cooperation (SDC), Germany's Kreditanstalt für Wieder-aufbau (KfW) and Bill and Melinda Gates Foundation (BMGF) through Foundation for Innovative and Novel Diagnostics (FIND) (Fig 1). The expanded medical activities are constituted of stocking rapid-diagnostic tests (RDTs) in a number of healthcare facilities and providing training for administration and referral, establishing additional laboratory capacity for parasitological confirmation of suspected cases, and the expansion of active screening with additional traditional teams as well as "mini-teams" on motorcycles. Moreover, since 2013, vector control (VC) has been supported by BMGF with the first deployment of Tiny Targets in 2014 following baseline surveys in 2013 (Fig 1). Since October 2016, the medical and entomological control activities have been integrated under the banner of the BMGF-funded Trypa-NO! Partnership [17].

## Strategies

**Strategy changes between 2014–2016.**   Prior to 2015, active screening (AS) was based on 12-person, 2-vehicle teams that would travel village-to-village while passive screening (PS) and parasitological diagnosis were only available at the Catholic Mission Hospital [22, 23]. Starting in 2015, the Catholic Mission Hospital was outfitted with loop-mediated isothermal amplification (LAMP), and a second hospital, the Bodo district hospital, was outfitted to diagnose, confirm, stage, and treat gHAT cases [18, 25]. Moreover, eight clinics were stocked with RDTs for screening, which then referred RDT-positive, suspected cases to the two hospitals for parasitological confirmation. In the following years, more facilities would be stocked, culminating in 31 clinics screening suspected cases in 2018. Eleven clinics were no longer stocked in 2019 in areas where no cases were detected (for more details, see Table E in S1 Text).

Starting in 2014, AS based on mobile units was complemented by smaller, "mini-mobile" teams riding motorbikes and administering diagnostic algorithms with RDTs [22, 23] (see Table D in S1 Text). While traditional teams confirmed cases onsite via microscopy exams of blood and cerebrospinal fluid, mini-mobile teams referred RDT-positive patients' samples to the two hospitals for additional screening with LAMP, in-person confirmation with parasitology, and treatment [18]. The screened villages have an average population of around 800 and 40–50% of the population present for screening by mobile teams. Since the structure of the teams is flexible between traditional and mini-mobile units, the teams can adapt to larger villages or higher participation rates (see Tables C-D in S1 Text). Transportation to the focus is limited during the rainy season of May to September due to the poor state of the roads and screening during the dry season is preferable as the population is less busy with agriculture [1, 22, 23]. Transportation is not much easier within the focus, and motorcycle teams have been designed to more easily reach villages and to provide transportation to suspected cases [23, 26].

To complement medical activities, in 2014 annual deployment of VC was initiated along the river and the marshland to control the tsetse population and interrupt transmission [20]. The deployment of these Tiny Targets—which attract tsetse and provide a lethal dose of insecticide through their impregnated mesh—has been very effective at reducing the local fly population density in Mandoul, with an estimated reduction of over 99% in only four months [20, 21] which has proved sustainable thereafter. The isolated nature of the Mandoul focus is believed to be one key reason for such a dramatic impact, although reductions of over 80% after one year have been found in various other regions across Africa using the Tiny Target approach [27–30]. High levels of vector reduction result in substantial disruption to the transmission cycle without the need for complete elimination of the tsetse. VC in Mandoul is further described by Rayaisse et al. [19].

**Table 1. Strategies against gHAT in the Mandoul focus.** Active screening (AS) coverage in the retrospective analysis is actual coverage in 2014–2019, and the recent mean coverage (2015–2019) thereafter, denoted as default coverage (ASd). Coverage in the prospective analysis is the recent mean (2015–2019, denoted as *Mean AS*) or the historical maximum for 2000–2019 (denoted as *Max AS*), which constitutes enhanced coverage (ASe). For passive screening (PS), PSd refers to default coverage, consistent with what was present before 2014 and PSe refers to enhanced coverage in clinics that were newly stocked with RDTs starting in 2015. Annually deployed vector control (VC) is expected to decrease the tsetse population by 99% in the first 4 months. Treatment is offered to all cases detected. The active screening algorithm specificity under *Mean AS & VC (a)* is 99.93% and for the strategies *Mean AS & VC (b)*, *Mean AS* and *Max AS* it is 100%. *Stop 2023 (No AS or VC)* signifies that AS and VC do not occur from 2023 onward. No P+/S+ indicates that both serological and confirmed cases must decline to 0 before scaleback. No P+ indicates that only cases confirmed by parasitology or trypanolysis must decline to 0 before scaleback. Serological confirmation consists of a person that is positive by CATT 1:8 dilution test, and parasitological confirmation is a person whose blood sample contains trypanosomes visible under a microscope.

| | Component Interventions | | | | | Scale-back criteria | |
|---|---|---|---|---|---|---|---|
| | **ASd** | **ASe** | **PSd** | **PSe** | **VC** | **AS & VC** | **PS** |
| *Retrospective analysis* | | | | | | | |
| Pre-2014 | ✓ | | ✓ | | | 3y no P+ | - |
| Imp. PS | ✓ | | ✓ | ✓ | | 3y no P+ | 5y no P+ |
| Addition of VC | ✓ | | ✓ | | ✓ | 3y no P+ | - |
| Imp. PS & VC | ✓ | | ✓ | ✓ | ✓ | 3y no P+ | 5y no P+ |
| *Prospective analysis* | | | | | | | |
| Mean AS & VC (a) | ✓ | | ✓ | ✓ | ✓ | 3y no S+ or P+ | 5y no S+ or P+ |
| Mean AS & VC (b) | ✓ | | ✓ | ✓ | ✓ | 3y no P+ | 5y no P+ |
| Mean AS | ✓ | | ✓ | ✓ | | 3y no P+ | 5y no P+ |
| Max AS | ✓ | ✓ | ✓ | ✓ | | 3y no P+ | 5y no P+ |
| Stop 2023 (No AS or VC) | | | ✓ | ✓ | | - | 5y no P+ |

**Counterfactual strategies for retrospective analysis and prospective analysis.** The strategies are found in Table 1 and illustrated in Fig 2 for both the retrospective and prospective analysis. In the retrospective analysis, we simulated the strategy that was actually implemented (*Imp. PS & VC*) to compare against strategies without one or both of the supplementary interventions. Alternative strategies included the interventions present before 2014 (*Pre-2014*, ASd

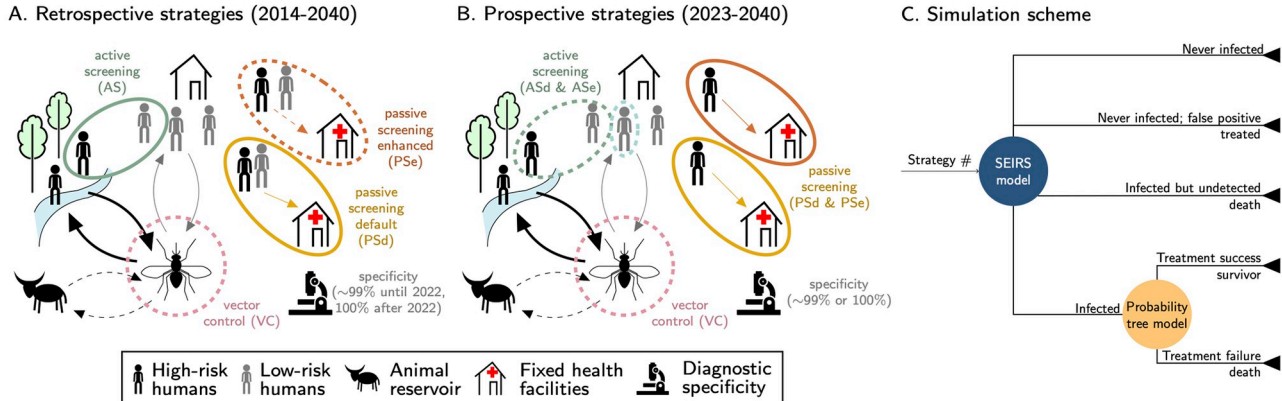

**Fig 2.** Schematic representation of the transmission models and the strategies in the (A) retrospective (2014 onward) and (B) prospective (2023 onward) analyses (adapted under CC-BY license from Rock *et. al* [31]). Solid lines in color represent intervention components that are present across strategies (e.g. active screening (AS) in the retrospective analysis and passive screening in the prospective analysis), whereas dashed lines in color are for intervention components only present in some strategies (e.g. vector control). Arrows in black denote transmission routes which include more transmission to high-risk individuals, and possible tsetse-animal-tsetse transmission pathways in some model variants (denoted by dashed black arrows). The imperfect specificity implies that there could be false positives that are treated. C) shows the modeling scheme which combines the transmission model (SEIRS) outputs with the probability tree outcomes. For the full diagram of the SEIRS model and the probability tree model of treatment, see Fig C in S1 Text).

and PSd and no VC), as well as strategies that either improved PS (*Imp. PS*) or only added VC (*Addition of VC*) to the Pre-2014 strategy. In all retrospective strategies, the same amount of AS took place and the passive detection rate was at least as much as before 2014.

In the prospective analysis, we look to alternative strategies that can be instituted going forwards starting in 2023 based on the actual situation in 2020 ("analytical present" in Fig 1). Therefore all these simulations include the intensified strategies that took place since 2014 combining AS, the expanded network of PS, and VC. All prospective strategies simulate AS in 2021 and 2022 at the mean coverage of 2014–2019 (19,628 persons per year), and changes to strategies are assumed to take place in 2023.

**Post-suppression phase.**   Suppression and post-elimination are defined here as the cessation of AS and VC after a pre-defined number of consecutive years of zero detected cases by any screening modality (AS or PS). After 2023, we simulated the impact of cessation of AS and VC if no cases are observed for a given period of time. For *Stop 2023 (No AS or VC)*, AS and VC stops immediately (it does not occur in 2023 onward). Under strategy *Mean AS & VC (a)*, AS and VC shut down when three years transpire of neither a serological positive (S+) nor a parasitologically confirmed (P+) case. For all other prospective strategies, AS and VC, when applicable, will shut down after three years of no detected P+ cases, even if S+ cases continue to be diagnosed and treated with fexinidazole. We rely on a realistic metric of disease suppression—either serological suspects or parasitologically confirmed cases—in order to simulate the risk and size of the resurgence if vertical interventions are ceased mistakenly early.

In the event that PS detects new P+ cases of either stage, an epidemiological investigation would take place to determine where the case was infected, if possible. In all strategies except for *Stop 2023 (No AS or VC)*, AS is assumed to be reactivated as reactive screening (RS) in the villages surrounding where the case was infected until there are no more P+ cases for one year. However, once VC stops it would not be restarted, even if more cases are found. The availability of RDT screening in health facilities (PS) is assumed to remain constant for five years after no P+ cases have been reported (and for five years after no S+ or P+ in the *Mean AS & VC (a)* strategy). After that, PS clinics are expected to cease stocking RDTs and PS would only take place in the two hospitals that have been providing screening, confirmation, and treatment (Catholic mission hospital and Bodo district hospital). However, no possibility of renewed expansion of RDT-stocked facilities was assumed. This is a reasonable assumption as only 0.1% of our simulations had further case reporting after five consecutive years with zero cases and no new cases arose.

Lastly, Chadian samples are shipped to Burkina Faso for Trypanolysis for post-treatment confirmation of S+ samples. Since only a few dozen samples are sent each year and there is no documentation of the cost to do this, we have omitted Trypanolysis costs from the analysis.

## Transmission and treatment models

To evaluate the health outcomes of different strategies against gHAT, we employed a previously published and validated dynamic, transmission "susceptible-exposed-infected-recovered-susceptible (SEIRS)" model [21] coupled here with a probability tree model of treatment outcomes, discussed in detail below (see Fig 2C and Fig C in S1 Text). A glossary of technical terms is found in S2 Text.

**Transmission model.**   We derived our simulations from an ensemble of models with compartments representing disease stages in humans and (possibly) non-human animals through tsetse transmission. The models are formalized through a system of ordinary differential equations (ODEs) fit to HAT Atlas data [32], and the ensemble is constituted of a sample of posterior predictions weighted by the model evidence for each structure. The model structure was

originally developed by Rock et al. [33], modified for the Chadian context in Mahamat et al. [20], and was recently updated by Rock et al. [21]. Finally, we provided another update here by revising our model fits to account for 3.15% annual population growth in the province of Mandoul [34], whereas a stable population was assumed previously [21]. Through numerical integration of the ODE system, we calculate the average expected dynamics. The activities that were actually implemented in Mandoul for the period of 2000–2020 are all represented in the model simulations, including variable active screening in part due to social insecurity and civil war, but also due to changes in financing. New infections and life years lost in stages 1 and 2 are deterministic outputs. Chance events, such as case detection and deaths, were simulated by sampling from a beta-binomial distribution with a rate parameter equal to the incidence of infected individuals who reported to care and are diagnosed in fixed facilities or attend AS by mobile teams and are detected.

An improvement in detection rates in fixed care facilities was fitted to the data, as reported in Rock et al. [21]. Both true and false positives are simulated, depending on the specificity and sensitivity of the algorithm in operation for the given year. Serological positives (S+) are defined as those cases that were positive by the card agglutination test for trypanosomiasis (CATT) 1:8 dilution, but for whom no trypanosomes were found in the blood. Parasitologically confirmed (P+) cases are those for whom trypanosomes can be microscopically visualised in blood. Patients who also have trypanosomes in their cerebrospinal fluid or an elevated white blood cell (WBC) count of more than 5 WBC/$\mu$l are classed as stage 2 cases, otherwise, they are classed as stage 1 cases. The remaining infected people who never report to screening, whether passive or active, are assumed to result in undetected gHAT deaths (see Section S1.2 in S1 Text, and especially S1.2.3).

The population actively screened by mobile units is fixed based on the data before 2019, and according to projected levels for different strategies starting in 2023. The active screening coverage for 2020–2022 was assumed to be the mean level for 2015–2019 (i.e. 19,628 people in each year). For passive screening, we assume in our projections that each infected person not picked up by active screening has the same average time to detection as infected people in 2015–2020 (a faster detection rate compared to the early 2000s). The key outputs of the transmission and detection model include deaths among infected people that were never detected, cases that were S+ or P+ confirmed, detected cases in stages 1 and 2, and disability-adjusted life-years (DALYs) before and after presenting to care for all interventions.

Tsetse control via Tiny Targets was explicitly simulated in the model to match strategies with VC in our retrospective analysis and in all strategies from 2014 in our prospective analysis. This is done by including an additional mortality term in the tsetse model equations linked to the probability of a tsetse finding and receiving a lethal dose of insecticide during the host-seeking phase of its gonotropic cycle. We assume that this probability is greatest directly after the new Tiny Target deployment and gradually decreases due to a range of factors (*e.g.* vegetation growth, loss due to rain) over the year. The value was fitted based on 2000–2019 case data and yielded a 99% reduction in tsetse after four months, in line with the reduction measured by tsetse traps density in the field [20, 21]. More details on modelling tsetse control in this 'Warwick gHAT model' framework are given elsewhere [21, 35]. Following cessation of VC (dictated by strategy choice) the tsetse population is allowed to recover in the model through the removal of the additional tsetse mortality term in the equations.

**Treatment model.** P+ cases are treated as stage 1 or 2 cases according to the presence of trypanosomes or elevated WBC count in cerebrospinal fluid (stage 2 for positive findings). Patients who were positive according to CATT with dilution 1:8 but who were not confirmed by parasitology have also been treated; these serological cases have been classified as stage 1 or 2 depending on symptoms [21].

The outcome of treatment is modelled according to a branching process formalized by a probability tree (see Fig 2C, and Fig C in S1 Text) using parameters summarized in Table 2 and discussed in detail in S4 Text. The probability of treatment failure and diagnosis determines convalescence and death outcomes. Before 2020, there were only two treatments: pentamidine for stage 1 cases and nifurtimox-eflornithine combination therapy (NECT) for stage 2 (see Fig 1) [2]. Starting in 2020, fexinidazole was available for inpatient and outpatient treatment of stages 1 and 2 for a subset of patients, as delineated by WHO recommendations (see Fig 1) [8]. Further details about the model are found in Section S1.4 in S1 Text and in Rock *et al* [21].

## Outcome metrics

The gHAT transmission model simulates an open cohort and produces annual outputs of cases in stage 1 and stage 2 (partitioned by whether they were identified in AS or PS), person-years spent with stage 1 and stage 2 disease, deaths from cases that were never reported, the number of new infections created, the number of people actively screened and whether or not vector control occurred. These were computed for each of the different analyses and strategies. Case outputs were used in turn in our probability tree model to determine which cases were given the different treatments (depending on stage and whether fexinidazole was yet available), and possibly second-line treatment based on the failure probability of first-line drugs. Additional deaths from failed treatments were also computed. Finally, to compute the DALY estimates under each strategy for each year, the total deaths (unreported and reported) were combined with the weighted person-years spent infected [58, 64]. Years of life lost posterior to the analytic horizon are counted in present-day terms by applying a 3% discounting rate.

The year of elimination of transmission (EoT) is defined in our deterministic model as the first year in which less than one new infection occurs, in line with the proxy threshold used to determine EoT in other gHAT modelling [65]. This level of new infections has been found to be an adequate approximation of the threshold to simulate local extinction as characterized in comparisons with stochastic models [66]. It should be noted that this may or may not be equivalent to the year when 0 new cases are detected, and because of the partially observed nature of the new cases, we use detected cases to determine the year of activity cessation. The probability of EoT by 2030 is the proportion of the 5000 iterations where the number of new cases—whether detected or undetected—reaches less than one case before or in the year 2030.

## Costs

The costs were calculated from the perspective of the healthcare payers collectively, including the government and all donors. A complete accounting of past costs for all interventions is nearly impossible because of the heterogeneous funding landscape for gHAT activities in Chad—a phenomenon that is present in many other gHAT-affected countries [67]. However, a detailed costing approach has been applied to VC, including by a contributor (APMS) [19], but screening and treatment activities are not specifically designated for gHAT in national health accounts in Chad. Therefore, although we have health-related data from the field, and limited resource-related data, we model the costs in order to come up with a holistic view of the costs involved in the endeavor from 2014. Costs are drawn from a variety of sources, from Chad whenever possible, and updated to 2020 values by taking into account the changes in the inflation rates as well as the exchange rates from local to US dollars (see Table 2), which we detail in the S4 Text.

Moreover, we model probabilistic uncertainty around how much costs would have been in different situations, based on combining our unit cost estimates with dynamic outputs from

**Table 2. Model parameters.** For further details and sources, see S4 Text. CIs: confidence intervals. AS & PS: active and passive screening, respectively, VC: vector control, PNLTHA: Programme de Lutte contre la Trypanosomie Humaine, NECT: nifurtimox-eflornithine combination therapy, CATT: card agglutination test for trypanosomiasis, S1 & S2: stage 1 & 2 disease, DALYs: disability-adjusted life-years, SAE: severe adverse events, RDT: Rapid Diagnostic Test.

| Variable Description | Statistical Distribution | Summary Mean (95% CIs) | Sources |
|---|---|---|---|
| **Screening parameters** | | | |
| Population | Fixed value | 41,000 | [20, 21]. See S4 Text, Section S4.5.1. |
| Population growth | Fixed value | 3.15% per year | [34]. See Section 1.2.2 in S1 Text & See S4 Text, Section S4.5.2. |
| PS: coverage of the population per facility | Fixed value for each year. | Retro: see S1 Text, Table E. Prosp: 100. | Trypa-NO records. See S4 Text, Section S4.5.11. |
| PS: number of facilities | Fixed value for each year. | See S1 Text Table E. Prospective, 22. | Trypa-NO records. See S4 Text, Section S4.5.13. |
| AS: coverage | Fixed value for each year. | See S1 Text Table C-D. | Trypa-NO records & HAT Atlas data. See S4 Text, Sections S4.5.3-S4.5.4. |
| AS: coverage for prospective strategies | Fixed values. | Mean AS: 19,628; Max AS: 22,146 | HAT Atlas data. See S4 Text, Section S4.5.7. |
| CATT algorithm: diagnostic specificity | Beta(31, 2) | 0.94 (0.84, 0.99) | [36]. See S4 Text, Section S4.5.9. |
| RDT algorithm: diagnostic sensitivity | Beta(230, 1) | 1.00 (0.98, 1.00) | [37]. See S4 Text, Section S4.5.8. |
| RDT algorithm: diagnostic specificity | Beta(3886, 24) | 0.99 (0.99, 1.00) | Trypa-NO records. See S1 Text Table E. |
| CATT algorithm: wastage during AS | Beta(8, 92) | 0.08 (0.03–0.14) | [38] See S4 Text, Section S4.5.15. |
| RDT algorithm: wastage during AS or PS | Beta(1, 99) | 0.01 (<0.01, 0.04) | [39] See S4 Text, Section S4.5.17. |
| CATT algorithm: wastage during PS | Beta(25, 75) | 0.25 (0.17, 0.34) | [39] See S4 Text, Section S4.5.16. |
| Pr. lost-to-follow-up, RDT + suspect | Year-specific. 2021 and later: Beta (21, 103) | See S4 Text, Table E. Prosp: 0.16 (0.11, 0.24). | See S4 Text, Table E and see Section S4.5.12. |
| **Screening cost parameters** | | | |
| AS capital costs (traditional team) | Gamma(200,000, 0.02) | 3192 (3178, 3206) | See S4 Text, Section S4.8.4. |
| AS management costs (traditional team) | Gamma(8.475–2167) | 18,361 (8308, 32,750) | See S4 Text, Section S4.8.5. |
| AS capital costs (motorcycle team) | Gamma(8.475, 277.1) | 2348 (1043, 4178) | See S4 Text, Section S4.8.1. |
| AS management costs (motorcycle team) | Gamma(70.05, 91.56) | 6412 (4951, 8018) | See S4 Text, Section S4.8.3. |
| AS followup costs (motorcycle team) | Gamma(70.05, 54.93) | 3843 (2992, 4782) | See S4 Text, Section S4.8.2. |
| CATT algorithm: cost per test used | Gamma(22.87, 0.02) | 0.45 (0.29, 0.67) | [38, 40–42]. See S4 Text, Section S4.8.6. |
| RDT algorithm: costs per test used | Gamma(8.475, 0.19) | 1.60 (0.71, 2.83) | [39] See S4 Text, Section S4.8.10. |
| Staging: lumbar puncture & lab exam | Gamma(2.42, 3.66) | 8.90 (1.45, 23.20) | [38, 41, 43]. See S4 Text, Section S4.8.7. |
| Confirmation: microscopy | Gamma(8.475, 1.27) | 10.64 (4.78, 18.68) | [39]. See S4 Text, Section S4.8.9. |
| PS capital costs of a facility (CATT + micro) | Gamma(8.475, 216.24) | 1826 (798, 3241) | [39] See S4 Text, Section S4.8.11. |
| PS capital costs of a facility (RDT only) | Gamma(8.475, 31.97) | 270.21 (119.69, 477.03) | [39]. See S4 Text, Section S4.8.13. |
| PS management costs (per focus, yearly) | Gamma(32.47, 12.18) | 396 (272, 543) | TrypaNO records. See S4 Text, Section S4.8.12. |
| **Treatment parameters** | | | |
| Proportion of cases age<6 | Beta(152.53, 2427.9) | 0.06 (0.05, 0.07) | [44, 45]. See S4 Text, Section S4.6.13. |

*(Continued)*

**Table 2.** (Continued)

| Variable Description | Statistical Distribution | Summary Mean (95% CIs) | Sources |
|---|---|---|---|
| Proportion of cases weight<35 kg & age>6 | Beta(8.3, 359.6) | 0.02 (<0.01, 0.04) | [45–54]. See S4 Text, Section S4.6.12. |
| Proportion of S2 cases that are severe | Beta(76.93, 44.87) | 0.63 (0.54, 0.72) | [8, 44–50]. See S4 Text, Section S4.6.1. |
| Length of treatment: pentamidine (days) | Fixed | 7 | [8]. See S4 Text, Section S4.6.2. |
| Length of hospital stay: NECT (days) | Fixed value | 10 | [8, 54]. See S4 Text, Section S4.6.4. |
| Length of hospital stay: fexinidazole (days) | Fixed value | 10 | [8, 54]. See S4 Text, Section S4.6.3. |
| Length of SAE (days) | Gamma (1.219, 2.377) | 2.89 (0.13, 9.94) | [55]. See S4 Text, Section S4.6.5. |
| Pr. of relapse: pentamidine | Beta(50.3, 665.48) | 0.07 (0.05, 0.09) | [44, 50–52, 56, 57]. See S4 Text, Section S4.6.6. |
| Pr. of relapse: NECT | Beta(15.87, 378.55) | 0.05 (0.02, 0.08) | [46–49, 53, 54]. See S4 Text, Section S4.6.7. |
| Pr. of relapse: fexinidazole | Beta(9.49, 496.54) | 0.02 (<0.01, 0.03) | [8]. See S4 Text, Section S4.6.8. |
| Pr. SAE: pentamidine treatment | Beta(1.43, 551.42) | 0.002 (<0.01, 0.01) | [44, 51, 52]. See S4 Text, Section S4.6.9. |
| Pr. SAE: NECT treatment | Beta(40.88, 367.8) | 0.05 (0.03, 0.08) | [46–49, 53, 54]. See S4 Text, Section S4.6.10. |
| Pr. SAE: fexinidazole treatment | Beta(3, 261) | 0.01 (<0.01, 0.03) | [54]. See S4 Text, Section S4.6.11. |
| **Treatment cost parameters** | | | |
| Hospital stay: cost per day | Gamma(5.45, 1.76) | 9.51 (3.27, 19.34) | [58–60]. See S4 Text, Section S4.9.1. |
| Outpatient consultation: cost | Uniform(2.48, 0.79) | 1.96 (0.32, 5.06) | [58–60]. See S4 Text, Section S4.9.2. |
| Course of pentamidine: cost | Gamma(100, 0.54) | 54.11 (44.12, 65.18) | [42]. See S4 Text, Section S4.9.3. |
| Course of NECT: cost | Gamma(100, 3.6) | 360.47 (291.82, 433.21) | [24]. See S4 Text, Section S4.9.4. |
| Course of fexinidazole: cost | Gamma(100, 2.2) | 220.0 (179.0, 265.1) | See S4 Text, Section S4.9.5. |
| Drug delivery mark-up | Unif(0.15, 0.25) | 0.20 (0.15, 0.25) | See S4 Text, Section S4.9.6. |
| **Vector control & cost parameters** | | | |
| Replacement rate of targets per year | Fixed value | 1 | [19]. |
| Deployment cost per year | Gamma(8.47, 81.32) | 68,847 (30,307, 123,000) | [19]. See S4 Text, Section S4.8.14. |
| **DALY parameters** | | | |
| Age of death from infection | Gamma(148, 0.18) | 26.63 (22.41, 31.08) | [44–56, 61]. See S4 Text, Section S4.7.4. |
| Average years lost at age of death | Interpolation of life expectancy at ages 20–35 | 45.4 (41.4–49.9) | [62]. |
| Disability weights: S1 disease | Beta(22.96, 147.21) | 0.14 (0.09, 0.19) | [63]. See S4 Text, Section S4.7.1. |
| Disability weights: S2 disease | Beta(18.37, 15.63) | 0.54 (0.37, 0.70) | [63]. See S4 Text, Section S4.7.2. |
| Disability weights: SAE | Uniform(0.04, 0.11) | 0.08 (0.04, 0.11) | [63]. See S4 Text, Section S4.7.3. |

the transmission model (number of people screened, number of vector control deployments) and probability tree model (numbers of different treatments given) (see S1 Text, Section S1.3). This approach is similar to that employed in another gHAT cost-effectiveness study for the Democratic Republic of the Congo [67], although each analysis is adapted to the country context. Cost components were broken down according to mean components and means of totals (since the sum of the means is not necessarily the mean of the sums). Final uncertainty was presented as the mean and the 95% predictive intervals (2.5th and 97.5th percentiles). The biggest drivers of uncertainty were determined by calculating the Expected Value of Perfect Partial Information—or the value to the program of reducing uncertainty to zero of any particular parameter. Due to the complex nature of the current model, the EVPPI was calculated by statistical approaches devised in previous literature [68].

## Cost-effectiveness analysis

Our cost-effectiveness analyses (one retrospective and one prospective) were performed from the perspective of the donors and the Ministry of Health in Chad combined, only accounting for direct costs (not indirect costs such as out-of-pocket expenses or days of lost work accrued by patients and their families). In each analysis, our default time horizon was until 2040 (2014–2040 for retrospective, and 2021–2040 for prospective) although shorter and longer time horizons were explored as sensitivity analyses. We selected 2040 as the default endpoint to better capture any benefits of EoT and/or cessation of activities—particularly during the 10 years following the goal year of 2030. This is one reason why we still used a default of 2040 for our time horizon in the retrospective analysis so that the assessment of cost-effective strategies is factored in medium-term benefits in the case that EoT occurs.

For each analysis, we selected one strategy to be our comparator strategy, representing a "status quo". For the retrospective analysis, the comparator (*Pre-2014*) is a counterfactual strategy representing the continuation of AS and PS activities without any additional improvements to PS nor introduction of VC. For the prospective analysis, the comparator is *Mean AS & VC (a)* which assumes that, from 2023, AS and VC continues at the average level until there are three years of no parasitologically (P+) or serologically (S+) confirmed cases, alongside continued annual deployment of VC and sustained PS at the 2019 level. Although horizons run until 2040, the activities might cease before that according to the cessation criteria set forth in Table 1.

Our main metric is the incremental cost-effectiveness ratio (ICER), defined as the additional costs incurred divided by the additional DALYs averted between one strategy and the next best strategy, taking out weakly or strongly dominated strategies [69]. Strategies or interventions are considered to be cost-effective if the ICER is less than a pre-defined willingness-to-pay threshold (WTP)—usually known as the cost-effectiveness threshold. In these calculations, both DALYs and costs were discounted at 3% per year.

Four mechanisms drive uncertainty in our modelling framework: the uncertainty in the parameterization of the dynamic model based on fitting to historical data, the stochastic nature of disease reporting and deaths, uncertainty surrounding treatment outcomes, which were parameterized based on literature based-estimates (see S4 Text). To account for all these different types of uncertainty, DALYs and costs are simulated by sampling 5,000 iterations for each of the alternative strategies in each analysis. Our outcomes are presented as means and 95% prediction intervals (bounded by the 2.5th and 97.5th percentiles of the simulated samples).

We use the net-benefit framework, which uses the net monetary benefits (NMB) as its key metric in order to account for parameter uncertainty in the cost-effectiveness analysis:

$$\text{NMB} := \text{WTP} \times \Delta\text{DALYs} - \Delta\text{Costs} \qquad (1)$$

which is a simple reworking of the ICER equation so that the NMB is positive when WTP is more than the ICER (ICER $= \frac{\Delta\text{Costs}}{\Delta\text{DALYs}} < $ WTP). Because we have run 5,000 Monte Carlo simulations, then we have 5,000 samples of NMB. Within this framework, the probability that a strategy is optimal is the proportion of samples for which the strategy yields the highest NMB for a given WTP, and the overall optimal strategy is the strategy that yields the maximum mean NMB.

We depict the uncertainty across a range of WTP values, as recommended by conventions, using a cost-effectiveness acceptability curve (CEAC) [70–72]. The CEAC shows the probability that each strategy is optimal in a line graph against increasing values of WTP while highlighting the optimal strategy at each WTP according to the mean NMB. WTP is challenging to enumerate precisely for any given country, as there are a range of financial and political

factors dictating how resources are allocated for specific disease programs. However, for context, the literature estimates have previously indicated that, in Chad, other disease programs have had WTPs of $30–518 per DALY averted [73, 74]. In 2020, the per-capita gross domestic product (GDP) in Chad was $659 [75], so based on previous WHO guidelines a strategy could have been considered to be "very cost-effective" at $659 per DALY averted or "cost-effective" at $1977 [58].

**Sensitivity analysis.** One scenario analysis was run to answer the question of whether our assessment of cost-effectiveness would look qualitatively different in a similar context to Mandoul but in the present, in which fexinidazole is available. For a calculation of the percentage of individuals who would be eligible for fexinidazole, see Table G and Table P in S1 Text.

Additional two-way sensitivity analyses were run. For the retrospective analysis, this included two shorter time horizons (a 10-year time horizon of 2014–2023, as well as a time horizons than end in 2030, 2040, and 2050). For the prospective analysis there was one shorter (2021–2030) and two longer (2021–2040 and 2021–2050) time horizons. Under each time horizon we produce results with and without discounted benefits and costs. These are available in the project website: https://hatmepp.warwick.ac.uk/MandoulCEA/v2 and well as in the supplementary information.

## Computational considerations

The transmission model was fitted and projections simulated using Matlab 2018b. Analyses using the probability tree, cost model and cost-effectiveness were performed using R version 4.1.1. Hardware needs are detailed in the accompanying Open Science Framework repository (https://osf.io/bjxwn/) which contains the code used to perform the analyses. For interested readers and policy-makers, we created a project website to showcase results and sensitivity analysis: https://hatmepp.warwick.ac.uk/MandoulCEA/v2.

## Results

The model has indicated that there was a quickly decreasing incidence of new infections in Mandoul from 2014 which preceded the accelerated decline in the number of cases reported (see Fig E in S1 Text). Moreover, the model also shows that infections, as well as cases reported, would have plateaued at a low level if *Pre-2014* strategy had remained in place, the decline was substantially accelerated by the additional activities of VC deployment and improved PS (see Fig E in S1 Text). Overall, after an initial increase in costs to set up improved PS and to start deployment of VC, costs showed a modest decrease from 2014–2020, but a more accelerated decrease is expected to occur in the next five years, as interruptions in transmission should allow the safe scale-back of vertical activities (Fig 3).

### Retrospective analysis

First, we ran a counterfactual scenario analysis to assess whether the previous increase in investment to gHAT activities in Mandoul represented a good use of resources (see Table 3).

**Transmission and reporting.** As outlined in our previously published transmission model analysis [21], strategies without VC would have had a very low probability of EoT by 2030: <1% if the strategies before 2014 had continued, or 6% if the network of RDT-based screening had been expanded (along with the motorcycle surveillance that we included) (see Table 3). However, with VC, the probability increases to >99% for both strategies, and we believe that EoT has occurred since 2016, and all subsequently detected cases were infected in 2015 or before.

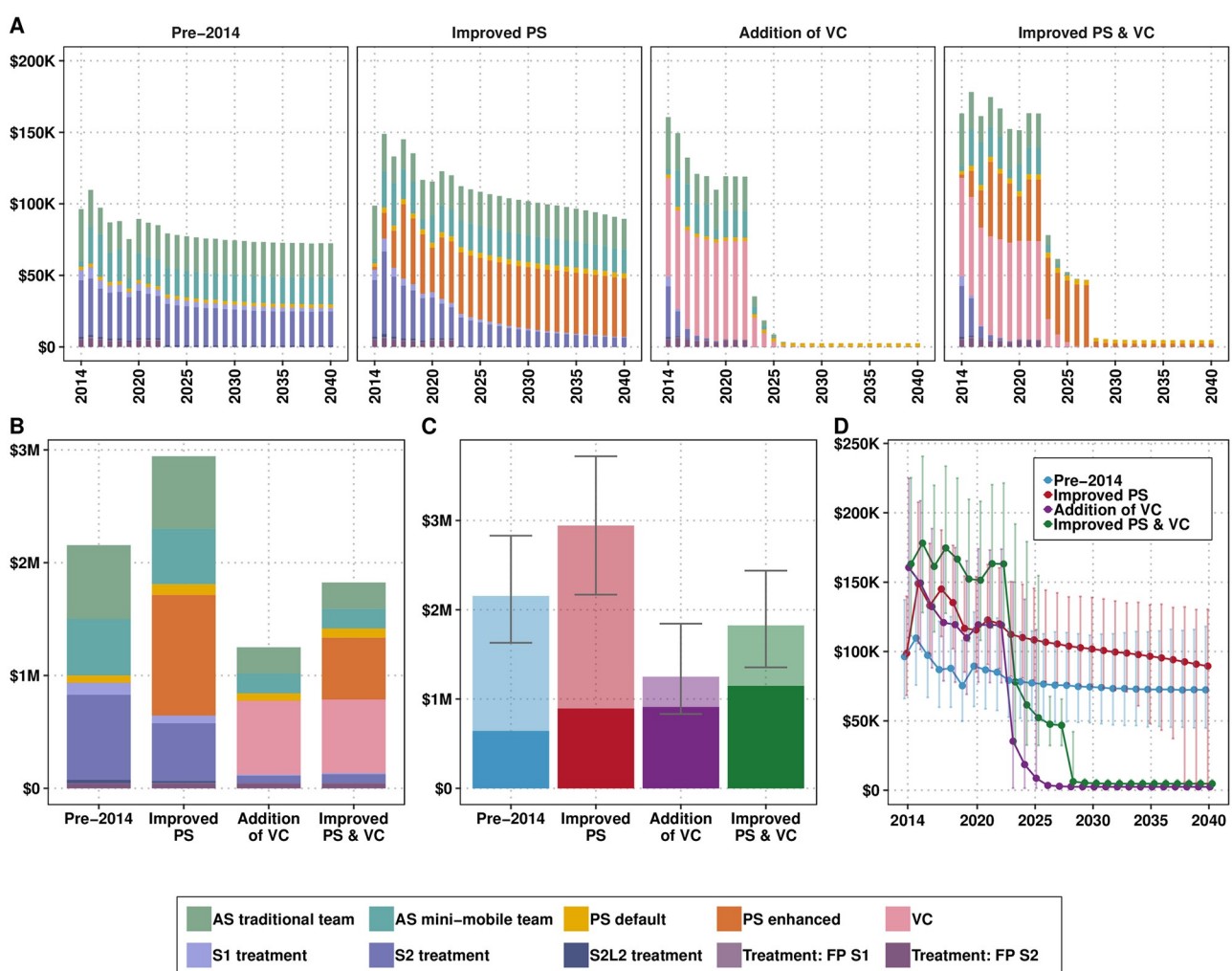

**Fig 3.** Costs by components of a strategy for the retrospective analysis: A) by year, B) for the period of 2014–2040, C) the costs for the period of 2014–2040 with uncertainty (transparent colors indicate the funds that have not been spent yet) and D) the total spent per strategy per year with 95% prediction intervals. See Table A and Fig B in S3 Text for total estimates and uncertainty of the costs spent between 2014–2020 compared to the whole time horizon (2014–2040). AS: active screening, PS: passive screening, VC: vector control, FP: false positive, S1: stage 1, S2: stage 2, S2L2: stage 2 rescue treatment.

The disease burden is substantially lower than it would have been with alternative strategies. In 2014, the strategies present at the time were on track to see an additional 2506 (95% PI: 1459, 3956) detected cases by 2040 or 1660 (95% PI: 787, 2959) detected if PS network had been improved (without the use of VC). The *Improved PS* strategy would have cut not just a third of cases, demonstrating the impact of PS on transmission dynamics, but more than halved deaths thanks to expanded screening and treatment: from 2160 (95% PI: 1062, 3689) to 943 (95% PI:160, 2444). VC strategies would have additionally cut by four-fifths the number of cases detected compared to the *Improved PS* strategy; adding VC to the 2014 strategies would have meant detecting only 213 (95% PI: 119, 337) cases or adding both VC and improved PS would have resulted in 238 (95% PI: 137, 363) detected cases. However, deaths, which come primarily from undetected cases, are fewer in the strategy that implements both improved PS clinics and VC (*Improved PS & VC,* 156 (95% PI: 72, 255)) than the strategy that detects fewer

**Table 3. Retrospective analysis.** Summary of effects, costs, elimination of transmission (EoT) by 2030, and cost-effectiveness with and without uncertainty for 2014–2040. Means are given along with 95% prediction intervals (PIs). YLL: years of life lost (to fatal disease), YLD: years of life lost to disability, DALYs: disability-adjusted life-years, PS: passive screening, AS: active screening, VC: vector control, ICER: incremental cost-effectiveness ratio, WTP: willingness to pay (USD per DALY averted), EoT: elimination of transmission.

| | Pre-2014 | Improved PS | Addition of VC | Improved PS & VC |
|---|---|---|---|---|
| **Health effects** | | | | |
| Reported cases | 2506 (1459, 3956) | 1660 (787, 2959) | 213 (119, 337) | 238 (137, 363) |
| Deaths undetected | 2160 (1062, 3689) | 943 (160, 2444) | 181 (108, 274) | 156 (72, 255) |
| Cases total | 4666 (2564, 7567) | 2603 (989, 5341) | 394 (253, 564) | 394 (248, 569) |
| Deaths detected[a] | 3 (1, 7) | 2 (1, 5) | 0 (0, 1) | 0 (0, 1) |
| YLD | 1711 (1000, 2767) | 798 (304, 1627) | 126 (77, 189) | 113 (62, 178) |
| YLL | 97,208 (48,106, 167,884) | 42,465 (7,153, 108,486) | 8,150 (4,814, 12,351) | 7,004 (3,222, 11,568) |
| DALYs | 98,919 (49,135, 170,561) | 43,264 (7,549, 110,191) | 8,276 (4,938, 12,485) | 7,117 (3,344, 11,707) |
| **Costs, in thousands US$** | | | | |
| AS costs | 1155 (863, 1512) | 1135 (798, 1502) | 409 (293, 574) | 408 (293, 566) |
| PS costs | 66 (38, 105) | 1165 (815, 1609) | 66 (38, 105) | 630 (454, 844) |
| VC costs | 0 (0, 0) | 0 (0, 0) | 654 (281, 1204) | 652 (281, 1190) |
| Treatment costs | 934 (531, 1509) | 644 (319, 1128) | 122 (77, 182) | 134 (85, 196) |
| Costs total | 2155 (1630, 2830) | 2943 (2169, 3719) | 1251 (833, 1844) | 1824 (1355, 2438) |
| **EoT** | | | | |
| Year of EoT | After 2050 | 2043 (2029, After 2050) | 2015 (2015, 2015) | 2015 (2015, 2015) |
| Prob EoT 2030 | <0.01 | <0.01 | >0.99 | >0.99 |
| **Cost-effectiveness without uncertainty (discounted)[b]** | | | | |
| DALYs averted | 0 | 20,722 | 35,653 | 36,264 |
| Cost difference | 0 | 572,679 | -439,958 | 18,268 |
| ICER | Dominated | Dominated | Min Cost | 749 |
| **Cost-effectiveness with uncertainty, conditional on WTP[c].** | | | | |
| WTP: $0 | 0.06 | 0 | 0.94(p) | 0 |
| WTP: $250 | 0 | 0 | 0.94(p) | 0.06 |
| WTP: $500 | 0 | 0 | 0.69(p) | 0.31 |
| WTP: $750 | 0 | 0 | 0.5 | 0.5(p) |
| WTP: $1000 | 0 | 0 | 0.39 | 0.61(p) |

[a] Detected deaths are those that occur due to treatment failure of loss-to-follow-up.

[b] Cost-effectiveness results are given for discounted DALYs and costs as per convention

[c] (p) is the preferred strategy; the strategy with the highest mean net monetary benefits

cases without improved PS (*Addition of VC*, 181 (95% PI: 108, 274)). Because DALYs primarily arise from fatal cases (see Table 3) the pattern on the impact of DALYs follows the pattern of deaths: continuing the *Pre-2014* strategy, would have brought on a loss of 98,919 (95% PI: 49,135, 170,561) DALYs in Mandoul, but the addition of improved PS and VC means that only 7117 (3344, 11,707) DALYs were incurred. The trend of DALYs across the years is illustrated in Fig A in S3 Text.

**Costs.** Over the 27-year period of 2014–2040, the total costs (without discounting) would have been highest had the *Improved PS* strategy been implemented (without VC) ($2.94M vs. $2.16M for the comparator), but the addition of VC without improved PS would have lowered the total costs ($1.25M), and the addition of both *Improved PS and VC* (actual strategy) is estimated to represent a very small increment in costs over the whole period ($1.82M). In 2015, it is estimated that costs increased by about 60% in order to institute the actual strategy conducted (*Improved PS & VC*) compared to the continuation strategy (*Pre-2014*).

The investment in VC and improved PS was partially offset by averted treatment costs over a 27-year horizon, as well as the substantially reduced long-running costs of AS activities (Fig 3 and Fig B in S3 Text). The costs of the strategy that was instituted (*Improved PS & VC*) were indeed higher for the period of 2014–2020, but the remaining costs in the strategy are the second lowest after the *Addition of VC* strategy (Fig 3 and Fig B in S3 Text). Specific estimates of the costs from 2014–2020 compared to the whole period can be found in Supplementary Information S3 Text, Table A.

**Cost-effectiveness.**   After taking a 3% discounting rate into account, the *Improved PS*, *Addition of VC*, and *Improved PS & VC* strategies would have averted 20,722 DALYs, 35,653 DALYs, and 36,264 DALYs, respectively. The strategy of *Improved PS* (and no VC) is *dominated*; in other words, while it costs more than any of the strategies with VC, it averts fewer DALYs. The investment of VC in addition to the continuation of the pre-2014 activities (*Addition of VC*) would have yielded cost savings by 2040, despite representing additional costs for the first few years. In fact, all strategies with VC would begin to yield cost savings in 2024 but will recover the investment completely by 2040.

Adding improved PS to the VC strategy (*Improved PS & VC*) costs an additional $456,214 (after taking into account 3% yearly discounting), for an additional 611 DALYs averted compared to *Addition of VC*; therefore, the ICER for the *Improved PS & VC* strategy is $749 per DALY averted. In short, while the investment in VC was cost-saving, the investment in improved PS was cost-effective at a willingness-to-pay of $749 per DALY averted (Table 3). At $500 per DALY averted, there is 31% probability that the implemented strategy was optimal, while at a WTP of $1000 per DALY averted, the probability that the strategy was optimal is 61% (see Fig 4).

## Prospective analysis

Going forward, imperfect test specificity in AS will incur direct costs in over-treatment, but those costs will be overshadowed by the inability to confidently cease vertical activities (i.e. VC and AS) (Fig 5B and Table 4). For all strategies with perfect diagnostic specificity, different combinations of interventions are predicted to make no difference in the number of cases detected (if there are any cases to be detected). The trend of DALYs across the years is illustrated in Fig C in S3 Text.

According to this analysis, any strategies with interventions other than basic continuation of PS are not cost-effective (Table 4 and in Fig D in S3 Text). What we conceived as this bare-minimum strategy (*Stop 2023 (No AS or VC)*) should cost around $399,000 ($282,000, $554,000) if vertical interventions are stopped in 2023, or at most $675,000 ($483,000, $1,008,000) for the period of 2021–2040 if VC and AS continues until no more parasitologically confirmed cases are detected (*Mean AS & VC (b)*).

## Scenario analyses

Different choices in times horizons or discounting of costs or disease burden would not have made a qualitative difference in our decision analysis, short of slight changes in the WTP values at which the *Improved PS & VC* strategy is cost-effective (see Fig E in S3 Text). Interested readers can explore the results of all scenario analysis on the project website https://hatmepp.warwick.ac.uk/MandoulCEA/v2/.

There was a difference in the results when we simulated treatment with fexinidazole for those patients who are eligible, but this difference would not change the optimal strategy substantially (see Table B in S3 Text). Fexinidazole treatment would have raised costs of treatment by 6%, or $8,000, since the cost of fexinidazole is higher than that of pentamidine

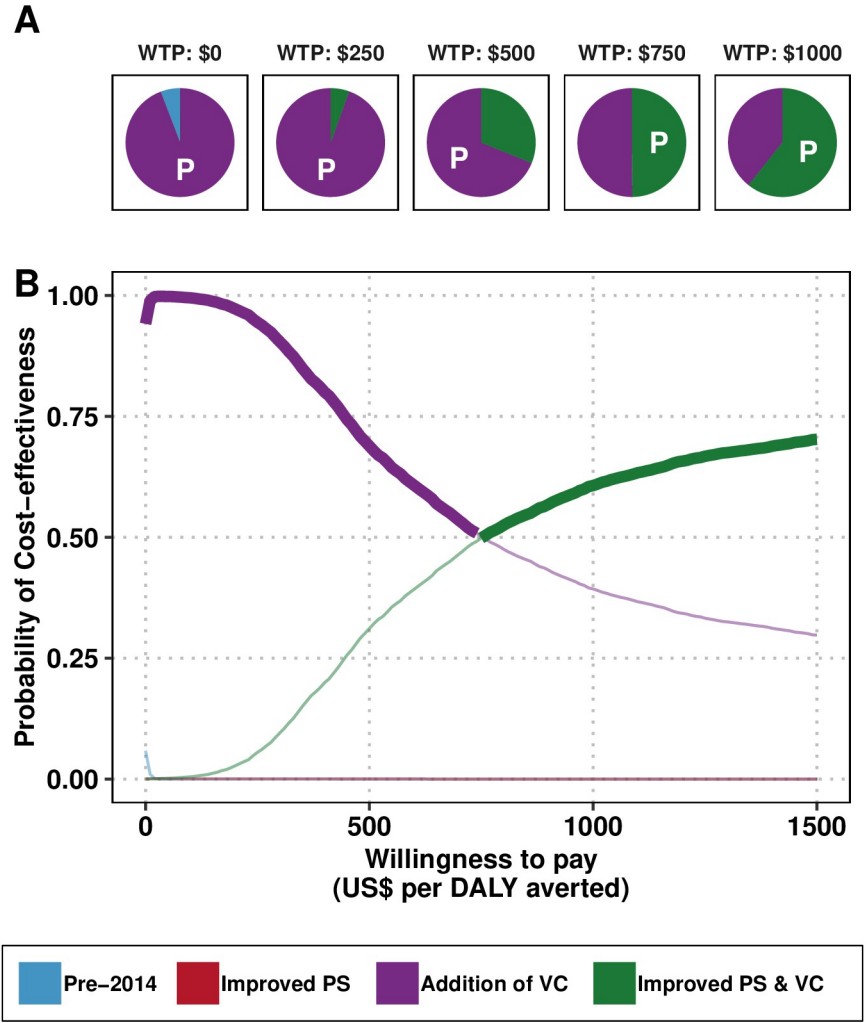

**Fig 4. Uncertainty in cost-effectiveness for the retrospective analysis.** A) Pie graphs depicting the probability that each strategy is optimal at the given willingness-to-pay (WTP) threshold. Strategies with the highest mean net monetary benefit are marked with a "P" for "preferred" strategy. B) Cost-effectiveness acceptability curves (CEACs) with the cost-effectiveness acceptability frontier (CEAFs) marked in bold. PS: passive screening, AS: active screening, VC: vector control, DALY: disability-adjusted life-years

although it is lower than that of NECT. However, the impact on total costs would have been an increase of $2,000, since treatment costs constitute a small part of all costs. The strategy *Improved PS & VC* would have an ICER of $746 per DALY averted instead of $749 per DALY averted.

## Uncertainty analysis

When we performed an uncertainty analysis of the factors that contribute the most to decision uncertainty (see Fig F in S3 Text), we found that the number of false positives considered stage 1 cases (S+ cases that would be P-) are the most influential parameter, followed by undetected deaths and the number of false positives considered stage 2 cases (S+ cases that would be P-). Other factors are the person-years of disease before detection of disease, the

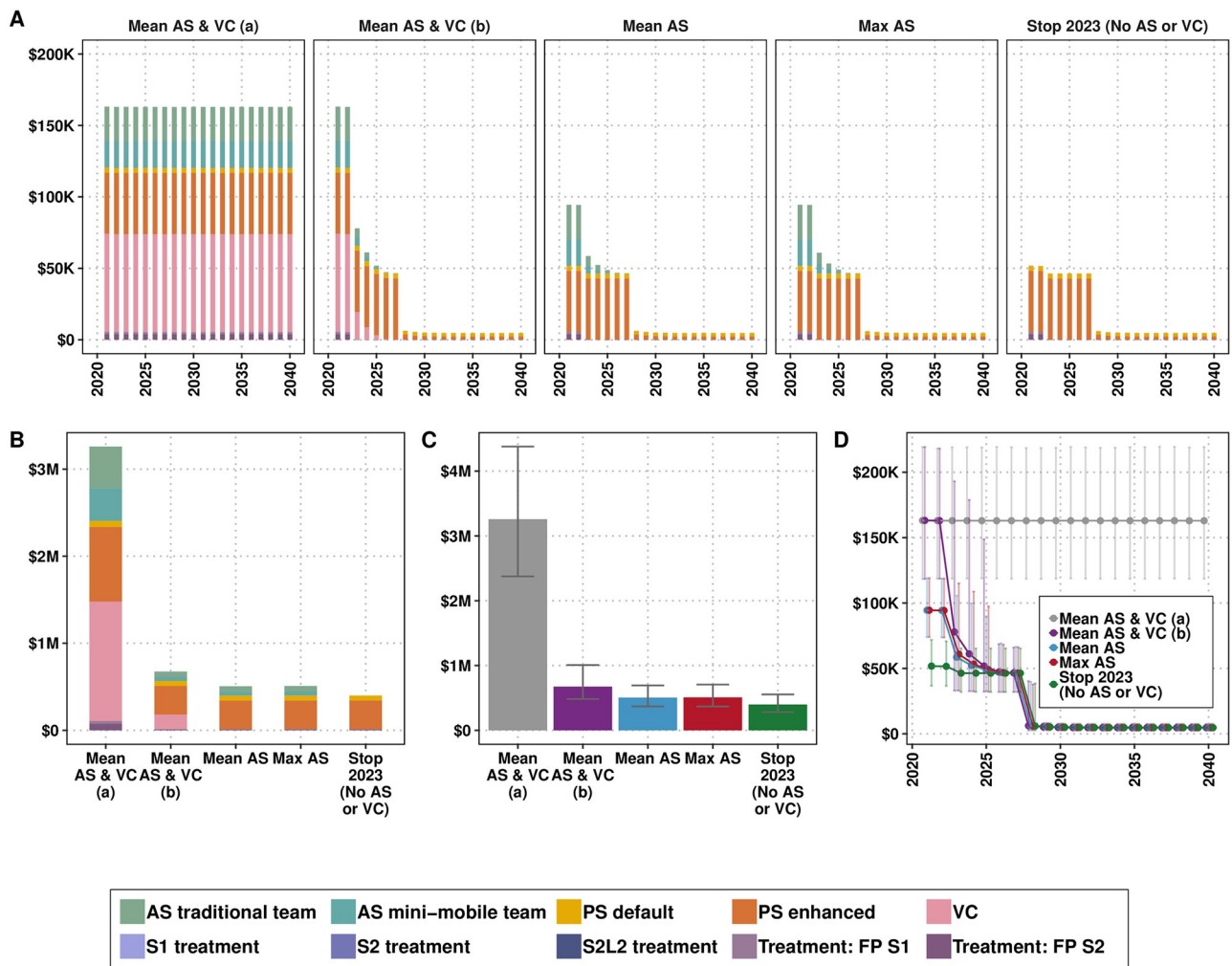

**Fig 5. Costs by components of a strategy for the prospective analysis.** A) by year, B) for the period of 2021–2040, C) the costs for the period of 2021–2040 with uncertainty, and D) the total spent per strategy per year with 95% prediction intervals. The specificity of the screening algorithm as part of the strategy *Mean AS & VC (a)* is 99.93% and for the strategies *Mean AS & VC (b)*, *Mean AS* and *Max AS* it is 100%. *Stop 2023 (No AS or VC)* signifies that AS and VC stop immediately (it does not occur in 2023 onward). Mean AS is the coverage of people screened for 2000–2019. Max AS is the maximum coverage of people screened for 2000–2019. AS: active screening, PS: passive screening, VC: vector control, FP: false positive, S1: stage 1, S2: stage 2, S2L2: stage 2 rescue treatment.

cost of microscopy, the capital cost of facilities that would be able to do screening only or screening and confirmation, the duration of severe adverse events, and the cost of treatment for an outpatient visits. Because there is no uncertainty as to the economically optimal strategy prospectively, we did not find it beneficial to perform an uncertainty analysis for the prospective strategies.

## Reporting guidelines

We completed two reporting guidelines are shown in S5 Text, CHEERS reporting guidelines for cost-effectiveness studies, and S6 Text, the PRIME-NTD criteria for modeling studies for policy.

**Table 4. Prospective analysis.** Summary of effects, costs, elimination of transmission (EoT) by 2030, and cost-effectiveness with and without uncertainty for 2021–2040. Means are given along with 95% prediction intervals (PIs). YLL: years of life lost (to fatal disease), YLD: years of life lost to disability, DALYs: disability-adjusted life-years, PS: passive screening, AS: active screening, VC: vector control, ICER: incremental cost-effectiveness ratio, WTP: willingness to pay (USD per DALY averted).

| | Mean AS & VC (a) | Mean AS & VC (b) | Mean AS | Max AS | Stop 2023 (No AS or VC) |
|---|---|---|---|---|---|
| **Health effects** | | | | | |
| Reported cases | 1 (0, 7) | 1 (0, 7) | 1 (0, 7) | 1 (0, 6) | 1 (0, 7) |
| Deaths undetected | 1 (0, 3) | 1 (0, 3) | 1 (0, 3) | 1 (0, 3) | 1 (0, 3) |
| Cases total | 1 (0, 8) | 1 (0, 8) | 1 (0, 8) | 1 (0, 7) | 1 (0, 8) |
| Deaths detected[a] | 0 (0, 0) | 0 (0, 0) | 0 (0, 0) | 0 (0, 0) | 0 (0, 0) |
| YLD | 3 (2, 5) | 1 (0, 1) | 1 (0, 1) | 1 (0, 1) | 1 (0, 1) |
| YLL | 26 (0, 133) | 26 (0, 133) | 26 (0, 134) | 26 (0, 133) | 26 (0, 134) |
| DALYs | 29 (2, 136) | 26 (0, 134) | 27 (0, 134) | 26 (0, 133) | 27 (0, 135) |
| **Costs, in thousands US$** | | | | | |
| AS costs | 854 (636, 1116) | 107 (65, 218) | 107 (66, 219) | 110 (65, 241) | 0 (0, 0) |
| PS costs | 927 (640, 1306) | 388 (272, 540) | 388 (272, 544) | 388 (271, 544) | 388 (272, 542) |
| VC costs | 1373 (602, 2448) | 169 (65, 394) | 0 (0, 0) | 0 (0, 0) | 0 (0, 0) |
| Treatment costs | 105 (66, 150) | 11 (5, 17) | 11 (5, 17) | 11 (5, 17) | 11 (5, 17) |
| Costs total | 3259 (2376, 4379) | 675 (483, 1008) | 506 (369, 693) | 509 (369, 707) | 399 (282, 554) |
| **Cost-effectiveness without uncertainty (discounted)[b]** | | | | | |
| DALYs averted | 0 | 2 | 2 | 2 | 2 |
| Cost difference | 0 | -1,876,594 | -2,041,277 | -2,038,178 | -2,145,409 |
| ICER | Dominated | 4,212,368 | Dominated | 1,216,989 | Min Cost |
| **Cost-effectiveness with uncertainty, conditional on WTP[c]** | | | | | |
| WTP: $0 | 0 | 0 | 0 | 0 | 1(p) |
| WTP: $250 | 0 | 0 | 0 | 0 | 1(p) |
| WTP: $500 | 0 | 0 | 0 | 0 | 1(p) |
| WTP: $750 | 0 | 0 | 0 | 0 | 1(p) |
| WTP: $1000 | 0 | 0 | 0 | 0 | 1(p) |

[a] Detected deaths are those that occur due to treatment failure of loss-to-follow-up.

[b] Cost-effectiveness results are given for discounted DALYs and costs as per convention

[c] (p) is the preferred strategy; the strategy with the highest mean net monetary benefits

## Discussion

Here, we present a case study of cost-effectiveness for intensified strategies to reach the EoT of gHAT. We evaluated the cost-effectiveness of the control program that was implemented in 2014 against less resource-intensive strategies. We showed that the addition of vector control to the *Pre-2014* activities was cost-saving, and the addition of both vector control and enhanced passive screening was cost-effective at $747 per DALY averted. Sensitivity analyses showed that even in the hypothetical scenario where it was possible to treat with the cheaper and easier-to-administer drug fexinidazole from 2014, our conclusions would have remained unchanged. Furthermore, our analysis suggests that going forward, it may be cost-effective to halt active screening and vector control as long as passive screening remains functional. Although active or reactive and passive screening could constitute both control and measurement activities, we do not believe active or reactive screening is necessary because in all strategies of the prospective analysis—with and without active screening—we expect fewer than 8 cases to be detected (Table 4).

This analysis features several novel properties over the only other past gHAT cost-effectiveness analysis in Chad [22]. That study, by Kohagne et al. examined the most effective way to

screen individuals in AS, but it didn't examine possible missed infections, nor modifications in other parts of the diagnostic or preventive cascade, like the addition of VC. Our study also differs from other studies in another respect: we employed a sophisticated, previously published model with parameters fitted to historical incidence data in Chad [20, 21] instead of using theoretical incidence levels [76]. This is the third time that this model—fit to local data—has been used in economic analyses in a specific setting [67, 77], and it is the first time it has been used outside of the Democratic Republic of the Congo—the country with the largest burden. One of the particular strengths of our analysis is that we used as much data as is available on the local transmission trends, economic costs, and resource use of gHAT interventions in Chad.

It should be noted that the addition of VC, in having such a quick and precipitous impact on transmission, also reduces the amount of uncertainty on burden and costs. For instance, the number of DALYs incurred by the *Improved PS* and *Addition of VC* strategies in the retrospective analysis have very similar lower bounds 4938 vs 7549, respectively, but the upper bounds were 110,191 vs 12,485, respectively. The uncertainty in costs takes longer to be reduced, but it is expected that the costs from 2021 on will be far less uncertain than those of counterfactual strategies without the supplementary interventions (see Fig 3).

The most important insight from our prospective analysis is the implicit costs of false positives (S+ but P- cases); if false positives compel the program to continue vertical interventions, then the costs of delayed cessation are much higher than the explicit cost of unnecessary treatment for people who might not truly be positive for the disease. We believe, therefore, that resources devoted to the prevention and treatment of gHAT in Mandoul could be diverted to address the existing burden in Moissala and Maro.

## Limitations

There are a few limitations to our analysis but we believe they do not obscure the quality of the conclusions. First, cost-effectiveness by itself does not address operational feasibility; for instance, having sufficient stock of RDTs throughout the world is not conveyed here unless there is an impact on prices as a reflection of pressures on supply. However, supply chain issues are beyond the scope of any cost-effectiveness analysis.

Second, it is critical to consider the data requirements necessary to verify EoT. It should be noted that our metric of elimination, i.e. no new infections, is a model output rather than an observable metric like cases reported, and a lack of cases reported may not be a good indicator of a lack of new transmission. It is therefore important to consider the impact of any delay or under-reporting of infections on verification of EoT. In addition, it is important to factor in data requirements to surveil against possible importation of infection from neighbouring endemic foci. It should be noted that in this analysis we do not explicitly model or cost activities aimed at verification of EoT, as the requirements for verification have not yet been published by WHO. Additional costs not included here include person-time required to compile an elimination dossier, although at present it is unclear whether and how much additional screening or quality assurance would be needed to provide evidence for meeting the requirements for elimination. Moreover, the criteria to continue monitoring in case of disease resurgence or to verify the country as having achieved EoT might require resources that are beyond the calculations in this paper. The paper intends to inform activities, but ultimately, national ownership of the strategies going forward is paramount for the success of the program.

Third, other foci in Chad contain important features different from those in Mandoul, including different ecological and epidemiological properties. Therefore we cannot extrapolate from this analysis that the same strategies which were cost-effective in Mandoul would also be cost-effective in Maro and Moissala. For instance, the tsetse populations in those regions are

not an 'ecological island', like in Mandoul, and Maro is contiguous with the Maitikoulou focus in the Central African Republic, thereby being at higher risk of disease re-introduction. It is, however, not only about transferring the activities but also about designing the activities based on local epidemiology and needs after considering the suitability of the region for the deployment of VC or the movements and needs of different communities at risk. However, the results in Mandoul suggest that short-term aggressive strategies aimed at elimination could represent an efficient use of resources. Maro and Moissala likely still need sustained vertical interventions to reach local EoT, but the analysis to determine which package of interventions is optimal is beyond the scope of the present study.

Fourth, there were a few simplifying assumptions on the structure of the model. This is a deterministic model, rather than a stochastic model which could capture chance events at low infection numbers, but based on this group's other work we would expect to see very similar dynamics between deterministic and stochastic structures [66]. Furthermore, we do not consider spatial connectivity and re-importations in this paper, but we point the interested reader to the related publication by Rock et al. [21] which indicates that resurgence is unlikely even if VC stops and tsetse populations recover quickly. We do not scale PS back up after scale-down to the two clinics that carry out both serological and parasitological confirmation even if cases are detected. However, resurgence after the scale-down of PS happened in only 5 out of 5,000 iterations. It should also be noted that the loss of expertise among staff members who rarely screen for gHAT cannot be sensibly accounted for in a model or a cost-effectiveness analysis. Lastly, this analysis does not include self-curing asymptomatic infections (see [78, 79] and Model Y in [80]).

## Conclusion

Our analysis shows that the expansion of screening activities and the introduction of vector control against gHAT after 2014 was helpful in eliminating transmission of the parasite in Mandoul as well as being a cost-effective use of resources. This result is interesting as it is not generally expected that strategies against relatively low-prevalence infections which are targeted for elimination can achieve this goal in an economically efficient manner. This indicates that the targeted use of supplemental interventions against gHAT in small hotspots of transmission has the potential to be cost-effective. Prospectively, we estimated the costs of a range of plausible strategies for Mandoul and found that scale-back of vertical activities would be economically sensible going forward, as long as national public health and governance stakeholders are satisfied that transmission has been halted.

## Supporting information

**S1 Text. An expansion of the materials and methods. Fig A: Remaining gHAT foci in Chad.** All remaining gHAT foci in Chad are located in the Southern region of the country. The exact extent of the area of transmission for Mandoul is hard to precisely define. The Mandoul focus was determined by geolocating all the gHAT cases indicated as living in Mandoul in the WHO HAT Atlas. Reprinted from Rock et al. [21] under a CC-BY license. **Table E:** Summary of passive screening in fixed health facilities. **Table D:** Summary of active screening activities by mini-mobile teams in motorcycles. **Table C:** Summary of active screening activities by traditional teams in trucks. Note: some of these cases might not be parasitologically confirmed. **S1.2 Text.** Additional method details covering the transmission model, strategy components, treatment model and health outcomes denominated as disability-adjusted life-years (DALYs). **S1.2.3 Text.** Additional method details about passive detection and unreported deaths. **Fig C: The transmission and treatment models** The transmission model,

depicting a Susceptible-Exposed-Infected-Recovered-Susceptible (SEIRS) model, which represents the progression of disease among low-risk humans (blue compartments), tsetse (purple compartment) and high-risk humans (red compartments), and the transmission of disease between the three groups. B) The probability tree representing treatment outcomes. Before 2020, a smaller tree was used constituting only the NECT and pentamidine branches for stage 1 and 2 disease, respectively, as fexinidazole was unavailable. Reproduced under CC-BY from Rock et. al [21] (part A) and Antillon et. al [67] (part B). **S1.4 Text.** Additional method details covering the treatment model. **S1.3 Text.** Additional method details covering the strategy components. **Table F:** Parameters for treatment eligibility. **Table G:** Eligibility for treatment. **Fig E: New 2000–2019 ensemble fit (incorporates 3.15% population growth).** The top row shows the number of people screened annually in the Mandoul focus from 2000–2019. Shaded regions denote vector control (VC) starting from 2014 (in blue) and improved passive screening (PS) starting from 2015 (in purple). The second and third rows show the active and passive case data as a solid black line, with grey-filled box and whisker plots denoting the median (centre line), 50% (box edges), and 95% (whiskers) credible intervals for the ensemble fit. Green box and whiskers (2019–2022) show the model projections, updated in the present study to reflect the known active screening levels. The tsetse reduction from 2014 and the passive detection rate improvement from 2015 were fitted. Inferred new infections each year are shown on the fourth line. This figure is the updated version of Fig S7 in Rock et al. [1], accounting for population growth. **Fig B:** A) Decision tree among four strategies for the retrospective analysis. B) A fifth branch is added for the prospective analysis, as there were 5 strategies to compare. For each strategy, the top two resulting branches of the model come from the transmission model only, and the bottom two resulting branches come from the treatment tree, which is fed outputs of detected cases from the transmission tree. **Fig D: Priors and posteriors for ensemble models fitted to 2000–2019 data for the old fit (no population growth) and the new fit (3.15% population growth).** The histograms show the ensemble model posteriors for the two different fits for each of the fitted parameters. The old fit (no population growth) is shown in blue and the 2000–2019 fit (3.15% population growth) is shown in orange. The black curve in each panel shows the prior distribution for that fitted parameter, it represents our belief before fitting was performed. **Fig F: New counterfactual strategy predictions 2014–2030 (incorporates 3.15% population growth).** The first and second rows show the active and passive case data as a solid black line, with grey-filled box and whisker plots denoting the median (centre line), 50% (box edges), and 95% (whiskers) credible intervals for the updated model fit (2000–2019). Counterfactual scenarios (CFSs) are shown from 2014 in other colours. Blue boxes denote the CFS in which no improvements to either vector control (VC) or passive screening (PS) were made, red boxes denote the CFS in which VC was not deployed but enhanced PS was started in 2015, and purple boxes denote the CFS where VC was deployed in 2014, but no enhanced PS was begun in 2015. From 2020 the projections are run under an assumption of mean active screening. The actual strategy switches from grey to green boxes from that year to reflect that it is a projection rather than a fitted. The third row displays the inferred new infections under each scenario, and the last row gives the computed probability of elimination of transmission (EoT) by each year for the different scenarios. This figure is the updated version of Fig S8 in Rock et al. [1] by accounting for population growth. **Fig G: Progression routes for humans once infectious.** The two blue boxes represent people infected in the first and second stages of gHAT respectively. The subscripts denote low-risk ($i = 1, 3$) and high-risk ($i = 2, 4$) sub-populations, some of whom never participate in active screening ($i = 3, 4$). Other disease state compartments are omitted here to focus on passive detection, however individuals enter from the exposed class "$E_{Hi}$" at rate $\sigma_H$. The gray box denotes individuals who may be actively detected and treated; the coverage of active screening varies from year to year and only people

in the "randomly participating" sub-populations may attend ($i = 1, 2$). If an individual is not actively detected, in stage 1 there is a rate of progression to stage 2 ($\varphi_H$) and a small rate of detection and treatment ($\eta_H(Y)$) which increases in 2015. Individuals in stage 1 move down each of these pathways proportional to the rates. If not detected actively in stage 2, individuals leave at a rate $\gamma_H(Y)$ which includes disease-induced deaths and passive detection and treatment. The rate of dying remains constant over time and so as the total exit rate $\gamma_H(Y)$ increases after 2015, a higher proportion of stage 2 people are detected and treated ($u(Y)$ becomes larger too). **Fig H: Passive detection rates over time** Left: $\eta_H(Y)$ denotes the rate of detection from the stage 1 compartment ($I_{1Hi}$). Right: $\gamma_H(Y)$ is the combined rate of exit from the stage 2 compartment ($I_{2Hi}$), which consists of either detection or death. **Fig I: Comparison of detected and undetected infections generated by the model and their relationship to active screening intensity.** A) detected cases and undetected cases. Box-and-whisker plots show the mean estimate, interquartile range, and 95% confidence interval of the total number of cases. B) the proportion of cases detected and the population tested by mobile screening teams. N.B. 0.5%, 4%, 20%, 44%, 68%, and 83% of 2017–2022 had zero true positive reports and zero deaths in the model simulations, and these simulations are excluded from the proportion calculations in the bottom panel. All results presented in this figure are taken from the model including stochastic sampling. **Table A:** Different model structures under consideration and their relative DIC scores for the fitting to 2000–2019 data. **Table B:** Different model structures under consideration and their relative DIC scores for the fitting to 2000–2019 data. **Table H:** Treatments and outcomes distributions for stage 1 and 2 patients, calculated according to the probability tree in B. SAE: severe adverse events. **Table I:** Active screening: cost function. **Table J:** Components of active screening costs. Full citations and explanations for the parameters will be given in SI Text 4. **Table K:** Cost breakdown for active screening activities. **Table L:** Passive screening: cost function. **Table M:** Components of passive screening costs. **Table N:** Cost breakdown for passive screening activities. **Table O:** Treatment: cost function. **Table P:** Parameters for treatment costs. Full citations and explanations for the parameters will be given in Supplementary Information S4 Text. **Table Q:** Cost per person for different gHAT treatments. Because these are costs averaged over all patients and SAEs are rare, the average cost per patient for SAE is low.
(PDF)

**S2 Text. Glossary of technical terms.**
(PDF)

**S3 Text. Additional results including figures and tables. Fig A: Disability-adjusted life-years (DALYs) accrued through time for different counterfactual strategies in the retrospective analysis. Table A: Costs spent vs total costs (in millions USD) for retrospective analysis.** Means are given along with 95% prediction intervels (PIs). **Fig B: Close-up of costs spent vs to spend in retrospective analysis.** A) Costs spent by intervention for each strategy in 2014–2020. B) Costs spent vs those to be spent by activity. See table A for total estimates and uncertainty in each period. **Fig C: Disability-adjusted life-years (DALYs) accrued through time for different counterfactual strategies in the prospective analysis. Fig D: Uncertainty in cost-effectiveness for prospective strategies.** A) Pie graphs depicting the probability that each strategy is optimal. Strategies with the highest mean net monetary benefit are marked with a "P". B) Cost-effectiveness acceptability curves (CEACs) with the cost-effectiveness acceptability frontier (CEAFs) marked in bold. **Fig E: Cost-effectiveness acceptability curves (CEACs) for retrospective strategies.** Cost-effectiveness acceptability curves (CEACs) for retrospective strategies, sensitivity analysis with alternative horizons with the cost-effectiveness acceptability frontier (CEAFs) marked in bold. **Fig F: Expected value of perfect partial**

**information (EVPPI) for retrospective analysis. Table B: Sensitivity analysis if fexinidazole had been available since 2014.** Summary of effects, costs, elimination of transmission (EoT) by 2030, and cost-effectiveness with and without uncertainty. Highlighted lines are the results that changed after simulating treatment with fexinidazole (compare to Table 3). Means are given along with 95% prediction intervals (PIs). YLL: years of life lost (to fatal disease), YLD: years of life lost to disability, DALYs: disability-adjusted life-years, PS: passive screening, AS: active screening, VC: vector control, ICER: incremental cost-effectiveness ratio, WTP: willingness to pay (USD per DALY averted), EoT: elimination of transmission.
(PDF)

**S4 Text. Model parameter glossary.**
(PDF)

**S5 Text. NTD-PRIME checklist.**
(PDF)

**S6 Text. CHEERS checklist.**
(PDF)

## Acknowledgments

The authors thank PNLTHA for original data collection, and the WHO for data access (in the framework of the WHO HAT Atlas [6, 14, 32]).

## Author Contributions

**Conceptualization:** Fabrizio Tediosi, Kat S. Rock.

**Data curation:** Marina Antillon, Paul R. Bessell, Alexandra P. M. Shaw, Iñaki Tirados, Philippe Solano, Severin Mbainda, Justin Darnas, Xia Wang-Steverding, Mallaye Peka.

**Formal analysis:** Marina Antillon, Ching-I Huang, Ronald E. Crump.

**Funding acquisition:** Sylvain Biéler, Fabrizio Tediosi, Kat S. Rock.

**Investigation:** Alexandra P. M. Shaw, Iñaki Tirados, Albert Picado, Sylvain Biéler.

**Methodology:** Marina Antillon, Ching-I Huang, Alexandra P. M. Shaw.

**Project administration:** Emily H. Crowley, Kat S. Rock.

**Software:** Marina Antillon, Ching-I Huang, Samuel A. Sutherland, Ronald E. Crump, Paul E. Brown, Kat S. Rock.

**Supervision:** Fabrizio Tediosi, Kat S. Rock.

**Validation:** Marina Antillon, Samuel A. Sutherland.

**Visualization:** Marina Antillon, Paul E. Brown.

**Writing – original draft:** Marina Antillon, Emily H. Crowley, Kat S. Rock.

**Writing – review & editing:** Marina Antillon, Ching-I Huang, Samuel A. Sutherland, Ronald E. Crump, Paul R. Bessell, Alexandra P. M. Shaw, Iñaki Tirados, Albert Picado, Sylvain Biéler, Paul E. Brown, Philippe Solano, Severin Mbainda, Justin Darnas, Xia Wang-Steverding, Emily H. Crowley, Mallaye Peka, Fabrizio Tediosi, Kat S. Rock.

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
