## [Decision Letter · Decision Letter 0]

29 Oct 2022

Dear Ms. Antillón,

Thank you very much for submitting your manuscript "Health economic evaluation of strategies to eliminate gambiense human African trypanosomiasis in the Mandoul disease focus of Chad" for consideration at PLOS Neglected Tropical Diseases. As with all papers reviewed by the journal, your manuscript was reviewed by members of the editorial board and by several independent reviewers. In light of the reviews (below this email), we would like to invite the resubmission of a significantly-revised version that takes into account the reviewers' comments. 

Major concerns are on consistency in the message the authors are projecting and whether the finding reflect a significant advancement of knowledge that the authors should address.

We cannot make any decision about publication until we have seen the revised manuscript and your response to the reviewers' comments. Your revised manuscript is also likely to be sent to reviewers for further evaluation.

Sincerely,

Paul O. Mireji, PhD

Academic Editor

Esther Schnettler

Section Editor

Major concerns are on consistency in the message the authors are projecting and whether the finding reflect a significant advancement of knowledge that the authors should address.

Reviewer's Responses to Questions

**Key Review Criteria Required for Acceptance?**

**Methods**

-Are the objectives of the study clearly articulated with a clear testable hypothesis stated?

-Is the study design appropriate to address the stated objectives?

-Is the population clearly described and appropriate for the hypothesis being tested?

-Is the sample size sufficient to ensure adequate power to address the hypothesis being tested?

-Were correct statistical analysis used to support conclusions?

-Are there concerns about ethical or regulatory requirements being met?

Reviewer #1: The objectives of the study a clearly stated and the design is well appropriate the address them. The methods used by the authors are well established and have previously used by the same team to address important HAT public health questions in different African countries. 

One of my concerns is the fact that the population was kept constant/stable over a 20 years projection period especially given that the analysis focuses on disease elimination. Even with a very low transmission rate, increasing population size may result in more cases than expected under a stable population, especially if population growth is significant in areas/villages of active transmission. Moreover, population increase may affect the number of iterations with resurgence. Though I don't expect it to affect the qualitative nature of the results, it may affect them quantitatively.

The second, which is very minor, is for the authors to clarify/provide references for their very rich model parameters. The authors did an impressive job gathering all these parameters values (e.g. Table 2), but I was enable to find supporting references for most of them. Maybe I missed them. But I think that readers will benefit of having all supporting references clearly provided in the main text.

Reviewer #2: Please see Comments under Summary & General Comments

**Results**

-Does the analysis presented match the analysis plan?

-Are the results clearly and completely presented?

-Are the figures (Tables, Images) of sufficient quality for clarity?

Reviewer #1: results are clearly and completely presented. But some Figures and Tables in the main Text are not properly referenced or not referenced at all. For example line 385, Table 3 is referenced but the exact reference seems to be Figure 3. Line 453 references Figure 2B, but it is not clear which figure it is. Line 415, please indicate that Figure A is from Supplementary Information S3. Lines 377 - 379 state that the model has indicated a quick decrease in incidence, but the authors provide no supporting evidence such as Figure.

Reviewer #2: Please see Comments under Summary & General Comments

**Conclusions**

-Are the conclusions supported by the data presented?

-Are the limitations of analysis clearly described?

-Do the authors discuss how these data can be helpful to advance our understanding of the topic under study?

-Is public health relevance addressed?

Reviewer #1: Conclusions are well written and supported by the presented data

Reviewer #2: Many conclusions are made and then the authors contradict themselves. Some of these conclusions are not a result of the modelling undertaken but authors own suggestions or propositions. Please see Comments under Summary & General Comments for more details.

**Editorial and Data Presentation Modifications?**

Reviewer #1: (No Response)

Reviewer #2: I think the paper would be better if the authors simply stick to conclusions based upon their Results of costs and cost effectiveness of the different strategies used in Mandoul to control gHAT. The 21 issues mentioned in Summary and General Comments need to be addressed.

**Summary and General Comments**

Reviewer #1: The manuscript presents a very interesting and timely analysis on the economic evaluation of control strategies for the elimination of gambiense human African trypanosomiasis. By focusing on the Mandoul disease focus which is the major HAT focus in Chad, the results presented here may be informative for other similar foci in Africa. The modeling study is well executed with appropriate sensitivity and uncertainty analyses, and the results are supported by the analyses. My minor comments are listed above.

Reviewer #2: In this study, the authors use a dynamic transmission model fit to epidemiological data from Mandoul to evaluate the cost effectiveness of combinations of active screening, improved passive screening and vector control activities (the deployment of Tiny Targets) aimed at interrupting transmission of gHAT to make recommendations on the best way forward based in terms of epidemiological projections and cost-effectiveness towards reaching the 2030 target of elimination as set by WHO.

GENERAL COMMENTS

1. a) The results of modelling, however, do not add much new knowledge to what is already known. Conclusions derived are very specific to Mandoul - an area with very low risk of reinvasion making it very difficult to extrapolate conclusions from this particular focus to other foci and caution is needed when drawing broader conclusions from such a very small and peculiar area. They suggest that VC did a very good job with value for money in this area and according to their calculations, the gains that have been made could be maintained by robust passive screening and AS and VC could now be stopped and could be considered for other foci in the country with active transmission (Page 2). This could be possible but in the very next sentence, they dilute their conclusion by saying “Our analysis speaks to comparative efficiency, and it does not take into account all possible considerations; for instance, any cessation of on-going active screening should first consider that strong surveillance activities will be critical to verify elimination of transmission and to protect against the possible importation of infection from neighbouring endemic foci”. Do we need complex modelling to conclude this? 

b) Another example of how the authors water down their own conclusions is provided on 

Pages 18 &19: Authors conclude that “we believe that the resources devoted to prevention and treatment of gHAT in Mandoul could now be diverted to address the existing burden in Moissala and Maro. Therefore, our study of the cost implications of the activities in Mandoul can serve as an illustrative guide for similar scale-ups of activities where appropriate in the effort to eliminate transmission in other persistently low-incidence locations of Sub-Saharan Africa.” 

Then on the same page 19 they tear down their own conclusion by saying “Other foci in Chad contain important features different to those in Mandoul including different ecological and epidemiological settings so therefore we cannot conclude that the same strategies that were cost-effective in Mandoul would be cost-effective in Maro and Moissala. For instance, the tsetse population in those regions are not an `ecological island', and Maro is contiguous with the Maitikoulou focus in the Central African Republic, thereby being at higher risk of disease re-introduction. Maro and Moissala likely still need sustained vertical interventions to reach local EoT, but that analysis is beyond the scope of the present study”. 

c) In 2017 (see reference 20), these authors through modelling concluded that a combination of case detection and treatment and vector control may have interrupted transmission and may lead to elimination in the Mandoul focus by 2020. Since then, what is the status in Mandoul? Has elimination been achieved as suggested by modelling at that time? The current conclusions need to be discussed within this context too. 

2. It is interesting that authors state in page 12 that “… as cases reported would have declined even if Pre-2014 strategy had remained in place, the decline was accelerated by the additional activities of VC deployment and improved PS. Overall, after an initial increase in costs to set up improved PS and to start deployment of VC, costs showed a modest decrease from 2014–2020, but a more accelerated decrease will occur in the next five years, as interruptions in transmission will allow the safe scale-back of vertical activities (Table 3)”. This could be a clear conclusion, in reality there are not so many differences between the different approaches, just when we project in the future. Moreover, this aspect can be biased by the calculation of undetected deaths (too high estimations?).

3. It also seems clear the importance of maintaining PS in the area after elimination. Maybe it would be most adequate to also keep the reactive screening (perhaps supplemented by reactive VC) more than just ceasing AS.

4. Use of DALYs to measure effectiveness is the usual approach but maybe it is not the best when facing a disease with very low prevalence in the way of elimination. In such a case, one needs to go to the beginning of interventions to estimate all the people non infected because of the application of the strategy of elimination. It requires establishing a starting point that is arbitrary. These authors have considered 2014 as starting point and 2040 as ending point but the situation could be very different if they are considering 2018 as starting point. DALYS are highly influenced by the number of deaths and there could be an overestimation of this factor.

5. The authors consider EoT when 0 cases are reached. This may be different from the WHO recommendation and view of partners that consider at least five years with 0 cases reported. 

OTHER COMMENTS:

6. Page 2: The authors mention “In the first year of renewed reports in Mandoul”. Could they be precise when exactly is that date?

7. Page 2: The authors refer to “a painful lumbar puncture for disease staging was required…”. Perhaps it is better to just mention “lumbar puncture…”, as pain is relative.

8. Page 3: The global definition of WHO goal for 2020 considers the time. So “High risk areas” should be defined as “more than one new reported case per year per 10,000 people averaged over the previous 5-year period”. The within-country indicator for HAT elimination as a public health problem also includes “over the previous 5-year period”.

9. Page 3: When talking about the 2014 previous situation the authors mention “after half a decade of low but unremitting case detection…”, it would be more exact to rephrase as “after a decade of case detection efforts, the number of cases showed a decreasing trend but reaching an unremitting low level of cases in the last years”

10. Page 3: Authors mention “We set out to perform a retrospective analysis for the Mandoul focus in which we look at what was done from 2014 compared to three less ambitious interventions. We examine what would have been the health economic implications if lesser strategies had been performed…”, What are these three less ambitions interventions? Similarly, what do they men by lesser strategies? 

11. Page 3: The authors mention that “they tested whether the introduction of fexinidazole in 2014 would have changed that selection”. This test is unclear as fexinidazole was far from being available in 2014 and it came to be used just in 2020. Hence, why to test its introduction in 2014? What are the advantages of fexinidazole with the previous treatment? It has to be considered that before 2014 all the cases were adequately treated.

12. Page 3: The comparison of Mandoul focus size with the area of Copenhagen or Austin (TX) is unclear. I’m afraid that the comparison is useless for many people as we are not familiar either with the area of Copenhagen and Austin and the use of these standards of size (Austin or Copenhagen) could be misinterpreted. Why not compare with other HAT foci?

13. Page 4: As far as I know and according to Chad HAT protocols, LAMP is not considered a confirmation test. So the statement “While traditional teams confirmed cases onsite via microscopy exams of blood and cerebrospinal fluid, mini-mobile teams referred RDT+ patients to the two hospitals for confirmation with parasitology or Loop-mediated Isothermal Amplification (LAMP)” should be amended accordingly. The definition of a case in Chad is stated later in page 8.

14. Page 5 and Fig 1: The authors refer to “less ambitious interventions” applied before 2014, but has the possible impact of some social factors presented in the past as social insecurity and civil war (2005-2010), which was present in the area before 2014 been considered? The AS activities in those years and therefore the number of cases detected was importantly related to the security conditions and this is not mentioned in the paper. This could have had an important impact. For scientists not to familiar reading this paper perhaps authors can clarify what these “less ambitious interventions” are!

15. Page 6 and Table 1: The case definition (S+ and P+) used in Chad is clarified in page 8 but it should be included before to understand table1.

16. Page 7: The authors mention that the cases not detected (unreported prevalence) “was assumed to result in undetected deaths”. Maybe this assumption is not correct. Some of these cases will result in undetected deaths but most probably many of them will be detected later when disease is most advanced and showing clear symptoms. This assumption can bias all the calculations done.

17. Page 13: The statement “The improvement of PS would have cut not just a third of cases, demonstrating the impact of PS on transmission dynamics, but more than halved deaths thanks to expanded treatment…” is unclear. How was treatment extended? The extension was mainly to diagnostic capacities through reinforcing PS and AS but the treatment remained the same and was not extended: Either before 2014 or after, “all the cases detected were treated”. Maybe in the model it would be good to know how many cases were detected in the new facilities opened when strategy was reinforced as some data show that cases were mainly detected in the facilities already working before the reinforcement of the activities. It would be also good to know what proportion of new cases were detected by “mini-teams” in the new reinforced approach. Could it be an impact of the improved security situation which allowed a better screening of populations? This should be clarified.

18. Page 14 and Table 3. The deaths undetected seem too high, probably related to the assumption that the cases detected in a certain moment result in deaths. This can importantly change the estimations.

19. Page 14. It is considered that VC and AS is continued until no more parasitological confirmed cases are detected. Usually, the strategies in use suggest continuing AS during at least three years after the last case is detected. Is this considered by the authors? As stated before “undetected deaths are the most influential parameter when uncertainty analysis.

20. Page 15. The authors say that the gains obtained could be maintained by robust passive screening. How do they define “Robust” passive screening? 

21. Page 19. While discussing the limitations of the study, the authors mention that “The paper is meant to inform activities, but ultimately, national ownership of the strategies going forward are paramount for the success of the program.” This needs to be clarified- what do they mean by “inform activities”? Secondly isn’t it common knowledge that national ownership of any control strategies is essential for sustainability of the control effort?

PLOS authors have the option to publish the peer review history of their article (what does this mean?). If published, this will include your full peer review and any attached files.

Reviewer #1: No

Reviewer #2: Yes: Dr. Rajinder Kumar Saini
---

## [Decision Letter · Decision Letter 1]

22 May 2023

Dear Ms. Antillón,

We are pleased to inform you that your manuscript 'Health economic evaluation of strategies to eliminate gambiense human African trypanosomiasis in the Mandoul disease focus of Chad' has been provisionally accepted for publication in PLOS Neglected Tropical Diseases.

Best regards,

Paul O. Mireji, PhD

Academic Editor

Esther Schnettler

Section Editor

<style type="text/css">p.p1 {margin: 0.0px 0.0px 0.0px 0.0px; line-height: 16.0px; font: 14.0px Arial; color: #323333; -webkit-text-stroke: #323333}span.s1 {font-kerning: none

</style>

Reviewer's Responses to Questions

**Key Review Criteria Required for Acceptance?**

**Methods**

-Are the objectives of the study clearly articulated with a clear testable hypothesis stated?

-Is the study design appropriate to address the stated objectives?

-Is the population clearly described and appropriate for the hypothesis being tested?

-Is the sample size sufficient to ensure adequate power to address the hypothesis being tested?

-Were correct statistical analysis used to support conclusions?

-Are there concerns about ethical or regulatory requirements being met?

Reviewer #1: (No Response)

Reviewer #2: see Summary & General Comments below

**Results**

-Does the analysis presented match the analysis plan?

-Are the results clearly and completely presented?

-Are the figures (Tables, Images) of sufficient quality for clarity?

Reviewer #1: (No Response)

Reviewer #2: see Summary & General Comments below

**Conclusions**

-Are the conclusions supported by the data presented?

-Are the limitations of analysis clearly described?

-Do the authors discuss how these data can be helpful to advance our understanding of the topic under study?

-Is public health relevance addressed?

Reviewer #1: (No Response)

Reviewer #2: see Summary & General Comments below

**Editorial and Data Presentation Modifications?**

Reviewer #1: (No Response)

Reviewer #2: see Summary & General Comments below.

**Summary and General Comments**

Reviewer #1: (No Response)

Reviewer #2: I am happy that the Authors have considered all the 24 points raised in my earlier review and have taken on board most of the points raised and amended the MS accordingly. I recommend the MS be accepted for publication.

PLOS authors have the option to publish the peer review history of their article (what does this mean?). If published, this will include your full peer review and any attached files.

Reviewer #1: No

Reviewer #2: **Yes: **Rajinder Kumar Saini

---

## [Editor Report · Acceptance letter]

19 Jul 2023

Dear Dr. Antillón,

We are delighted to inform you that your manuscript, "Health economic evaluation of strategies to eliminate gambiense human African trypanosomiasis in the Mandoul disease focus of Chad," has been formally accepted for publication in PLOS Neglected Tropical Diseases.

Best regards,

Shaden Kamhawi

co-Editor-in-Chief

Paul Brindley

co-Editor-in-Chief
